

# A hybrid model for streamflow prediction addressing spatial connectivity and non-stationary dynamics with adaptive graph learning and multiscale decomposition

Yueming Nan[1], Lizhi Tao[1,2*], Dong Yang[3], Haibo Zou[1,2], Yufeng He[1,2], Zhichao Cui [1] and Yuanbo Luo[4]

[1] Key Laboratory of Poyang Lake Wetland and Watershed Research of Ministry of Education & School of Geography and Environmental, Jiangxi Normal University, Nanchang, 330022, China

[2] Key Laboratory of Natural Disaster Monitoring, Early Warning and Assessment of Jiangxi Province, Jiangxi Normal University, Nanchang, 330022, China

[3] Jiangxi Provincial Key Laboratory of Environmental Pollution Control, Jiangxi Academy of Eco-Environmental Sciences and Planning, Nanchang, China

[4] Southern Marine Science and Engineering Guangdong Laboratory (Guangzhou), Guangzhou, China

*Corresponding author address: Lizhi Tao, Key Laboratory of Poyang Lake Wetland and Watershed Research of Ministry of Education & School of Geography and Environmental, Jiangxi Normal University, Nanchang, 330022, China. Tel: (86) 13117310498. E-mail address: taolz@jxnu.edu.cn*

**Abstract.** Accurate streamflow forecasting remains a challenge due to the pronounced nonlinearity and multiscale variability inherent in hydrological processes. In this paper, a hybrid logarithmically transformed complete ensemble empirical mode decomposition with adaptive noise (CEEMDAN)-based the spatial graph gated recurrent unit with adaptive graph structure (LCEEMDAN-ASGGRU) model is proposed to improve streamflow forecasting. The hybrid model is validated by forecasting daily streamflow at 14 stations in the Poyang Lake basin, a region characterized by complex river-lake interactions and significant spatial variability in streamflow magnitudes among stations. Results demonstrate that the LCEEMDAN-ASGGRU model shows superior predictive accuracy compared to benchmark models, achieving a mean Nash–Sutcliffe efficiency coefficient of 0.888 and mean root mean squared error of 264. The adaptive graph structure is spatially interpretable, closely aligning with known hydrological flow paths, while simultaneously capturing temporal similarity patterns among stations. In addition, a hidden Markov model with Gaussian Mixture Regression is used to quantify predictive uncertainty. Compared with other models, LCEEMDAN-ASGGRU yields the most reliable forecasts. This study demonstrates the effectiveness of coupling logarithmic transformation, CEEMDAN decomposition, and adaptive graph learning with graph neural networks, providing a novel integrated approach for improving streamflow forecasting accuracy under complex hydrological conditions.

Keywords: Streamflow forecasting; Graph neural network; Adaptive graph structure; CEEMDAN; Uncertainty quantification.

## 1 Introduction

Accurate and reliable streamflow prediction plays a crucial role in managing water resources, preventing floods, and conserving ecological systems (El-Shafie et al., 2007; Frame et al., 2022; Nearing et al., 2024). However, streamflow prediction



remains highly challenging due to the inherent complexity of hydrological processes, characterized by strong nonlinearity and non-stationarity, and influenced by numerous external factors (Tao et al., 2025).

Existing streamflow forecasting methods can be classified into two main types: process-driven and data-driven (Liu et al., 2022). Process-driven methods, also known as physically-based models, aim to build detailed and simulated watershed models using the underlying physics of hydrological processes (Gao et al., 2020). These models enable more realistic model structures, parameterizations, and calibrations (Gao et al., 2017). However, they require detailed parameterizations and extensive data inputs, and often encounter substantial uncertainties due to oversimplifications of the complex hydrological processes (Xiang and Demir, 2020). Compared with process-driven models, data-driven models can achieve a higher predictive accuracy with lower modeling requirements. Statistical models, as a classical subset of data-driven methods, have been extensively applied to streamflow forecasting in diverse hydrological regions, with representative models including autoregressive (Myronidis et al., 2018) and autoregressive integrated moving average models (Wen et al., 2019). However, due to their reliance on a linear assumption, conventional statistical methods struggle to represent the intricate nonlinear dynamics embedded within streamflow series, often resulting in low predictive performance.

Recently, machine learning has received considerable attention because of its potent learning capabilities and adeptness at handling complex nonlinear processes (Ni et al., 2020). Techniques such as support vector machine (SVM), random forest (RF), and artificial neural network (ANN) have shown improved performance in various hydrological prediction tasks (Adnan et al., 2020; Oppel and Schumann, 2020; Tan et al., 2018; Yu et al., 2023). Li et al. (2016) developed an RF model for forecasting lake water levels and demonstrated its superior accuracy over traditional models. Noori and Kalin. (2016) developed an ANN with a quasi-distributed watershed for daily streamflow forecasting, demonstrating that coupling ANN with semi-distributed models can lead to an improvement in daily streamflow prediction in ungauged watersheds.

Although conventional (shallow) machine learning models have proven effective for nonlinear streamflow prediction, their flat architectures limit their ability to learn the hierarchical representations required as hydrological datasets grow in size and complexity. This challenge has been effectively addressed by the rapid advancement of deep learning techniques and the acceleration of GPU computing, which together enable more efficient hierarchical feature extraction from large and complex datasets. The hierarchical structure of deep neural networks, featuring multiple layers of interconnected neurons, provides a greater ability to represent complex functions than machine learning methods (Quilty et al., 2022; Tao et al., 2023). Recurrent neural network (RNN) is a type of sequence model that maintains a vector of hidden states that propagates over time (Tao et al., 2021). However, conventional RNNs struggle to learn from long sequences, leading to the vanishing gradient problem (Ni et al., 2020). An improved version of the RNN, called the long short-term memory network (LSTM), offers unique advantages while maintaining the general characteristics of the RNN (Lin et al., 2021). Xiang et al. (2020) applied LSTM and sequence-to-sequence models for hourly rainfall-runoff prediction and showed that the combined architecture significantly enhances short-term flood forecasting accuracy. As a streamlined variant of LSTM, the gated recurrent unit (GRU) achieves comparable predictive performance to LSTM while reducing model complexity and parameter count, thereby improving computational efficiency in practical applications (Cho et al., 2014). However, LSTM and GRU are primarily designed for one-dimensional



sequences and therefore cannot fully exploit the spatial dependencies embedded in geospatial hydrological systems (Jin et al., 2024; Liu et al., 2023, 2022; Zhao et al., 2024).

Graph neural networks (GNNs), owing to their ability to operate directly on graph-structured data, have emerged as a promising approach for capturing both spatial heterogeneity and temporal dynamics embedded in these increasingly unstructured hydrological datasets (Bronstein et al., 2017). Graph-based learning has been a significant application area across various domains, including computer vision, natural language processing, recommender systems, chemistry, and transportation (Pradhyumna and Shreya, 2021; Rahmani et al., 2023; Reiser et al., 2022; L. Wu et al., 2023a, b). In hydrological forecasting, Sun et al. (2021) explored the application of several state-of-the-art GNNs, including GraphWaveNet, Graph Convolutional Network, and ChebNet, for streamflow forecasting in both gauged and ungauged basins. The results demonstrated that GNNs are effective in capturing spatiotemporal dependencies and offer a promising framework for large-scale, end-to-end streamflow prediction. A key determinant of GNN performance lies in how the graph structure is defined, as it directly affects the propagation and aggregation of spatial information throughout the network (Zhu et al., 2021). Most existing hydrological studies construct graph structures based on predefined spatial criteria, such as river connectivity or geographic proximity, which are typically referred to as static graphs (Gai et al., 2023; Jin et al., 2024; Lin et al., 2021; Liu et al., 2023). Based on basin topology, Liu et al. (2022) developed a directed graph deep neural network and achieved strong performance in multi-step runoff forecasting. Gai et al. (2023) designed and evaluated multiple static graph structures, including complete, information flow, and groundwater flow field association graphs, to better capture the spatial dependencies inherent in karst hydrological systems. However, two challenges still prevent accurate streamflow forecasting using GNN models.

The primary challenge lies in how to accurately define the spatial relationships among hydrological stations, which fundamentally determines how information is propagated across the graph. While recent studies have validated the potential of graph-based approaches in hydrological applications (Liu et al., 2023), these models still rely on predefined static graphs that cannot adapt to the underlying complex and evolving inter-station dependencies. They are typically developed for specific basins and calibrated to particular hydrological settings, limiting their generalization ability across regions or under changing environmental conditions. To overcome this limitation, Wu et al. (2019) first proposed GraphWaveNet, which learns an adaptive graph structure from traffic data to better capture hidden spatial dependencies. However, the application of this model to hydrology requires further exploration.

The second challenge lies in the intrinsic complexity of hydrometeorological data, characterized by strong nonlinearity and non-stationarity (Šraj et al., 2016). Directly analyzing original time series would ignore some features of different time scales. To overcome this limitation, hybrid approaches that combine deep learning with signal decomposition techniques have been increasingly applied to extract multi-scale patterns and improve model robustness (Adamowski and Sun, 2010; Karran et al., 2014). In particular, the complete ensemble empirical mode decomposition with adaptive noise (CEEMDAN) has been widely used to process non-stationary time series by decomposing them into a set of intrinsic mode functions (IMFs) with distinct frequency characteristics (Cao et al., 2019; Karijadi et al., 2023; Zeng et al., 2023). Zeng et al. (2023) developed a CEEMDAN-based model that integrates signal decomposition, reconstruction, and ensemble prediction, achieving high accuracy and



robustness in daily streamflow forecasting. These results demonstrate that CEEMDAN can effectively preprocess non-stationary streamflow data, thereby improving predictive performance.

In addition, quantifying forecast uncertainty is crucial to assess the predictive reliability of the model. Rather than incorporating uncertainty modeling into the prediction model, researchers have developed post-processing techniques to evaluate the predictive confidence of models. Representative methods include Bayesian model averaging (Duan et al., 2007), lower upper

bound estimation (Khosravi et al., 2010), Gaussian process regression (Sun et al., 2014), and hidden Markov model (HMM) with Gaussian Mixture Regression (GMR) (Liu et al., 2018).

In this study, we propose a logarithmically transformed CEEMDAN-based the spatial graph gated recurrent unit (SGGRU) with adaptive graph structure (LCEEMDAN-ASGGRU) for daily streamflow forecasting. The key contributions of this study are summarized as follows:

(1)    The LCEEMDAN-ASGGRU model is proposed for daily streamflow forecasting. Different from the existing models, the proposed model combines signal processing and a spatiotemporal model with adaptive graph learning, which captures multi-scale temporal patterns and dynamic spatial dependencies.

(2)    HMM-GMR is applied to quantify the forecasting uncertainty. Evaluation demonstrates that our proposed model achieves superior deterministic accuracy, especially under extreme flow conditions.

(3)    The hybrid LCEEMDAN-ASGGRU is applied to the Poyang Lake basin and compared with five models (LSTM, DTWSGGRU, FDSGGRU, ASGGRU, CEEMDAN-ASGGRU). The results show that the proposed model outperforms all five other models in terms of RMSE and NSE.



## 2 Data and problem formulation

### 2.1 Study area and dataset

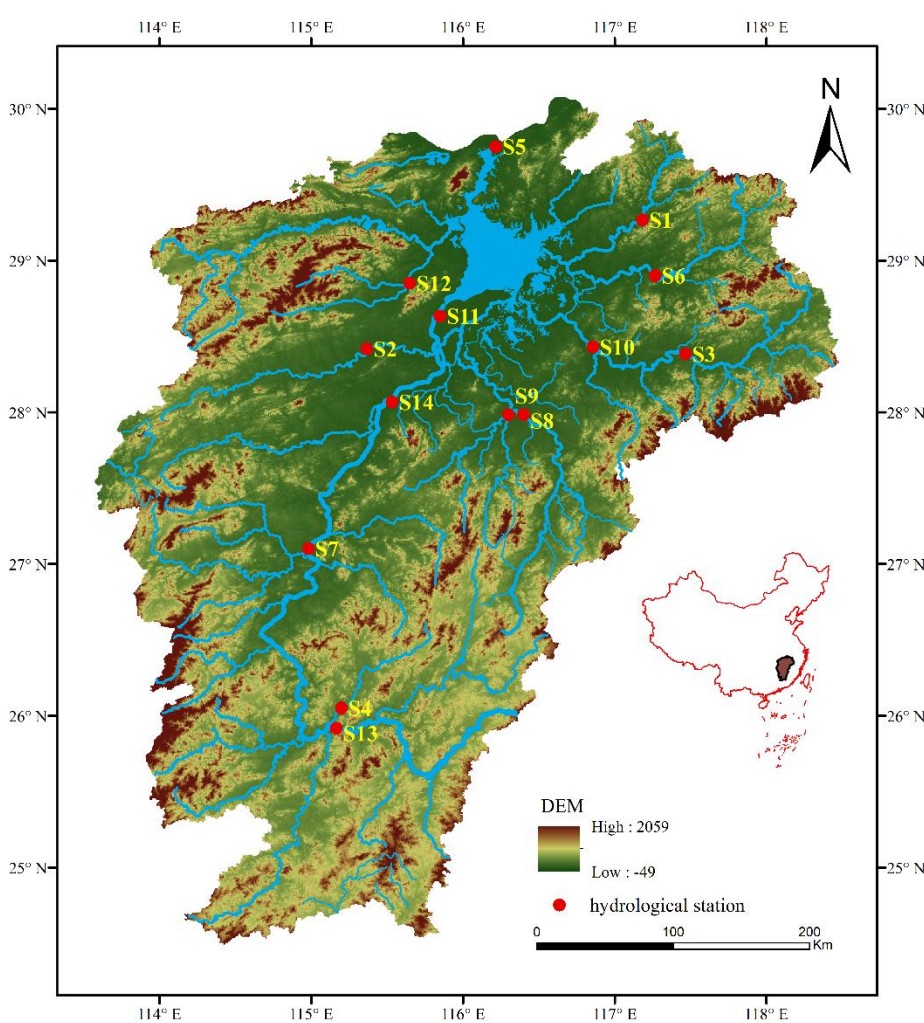

**Figure 1: Location of hydrological stations.**

The Poyang Lake basin is located in southeastern China, on the south shore of the middle and lower reaches of the Yangtze River. The basin is primarily fed by five major rivers—Ganjiang, Fuhe, Raohe, Xinjiang, and Xiushui—each with multiple tributaries and distinct hydrological characteristics. These rivers converge into Poyang Lake before eventually draining into the Yangtze River.

The dataset utilized in this study, obtained from the Hydrological Yearbook of China, consists of daily observations of



streamflow, temperature, and precipitation collected from 14 hydrological stations in the Poyang Lake basin. Figure 1 shows
the spatial distribution of selected stations, covering the five major river systems that drain into the Poyang Lake. The daily
data span a 12-year period (2009-2020), comprising a total of 4383 daily records. The first 80% of the data (January 1, 2009
to March 31, 2018) were used for training, while the remaining 20% (April 1, 2018 to December 31, 2020) were reserved for
testing.

### 2.2 Graph construction strategies

In graph-based hydrological modeling, the spatial relationships among observation stations are encoded as graphs to capture
the dependencies arising from geographical proximity, hydrological flow paths, or behavioral similarity (Sun et al., 2021). The
underlying graph structure serves as the foundation for message propagation in GNNs, and therefore has a significant impact
on model performance and generalization.

To formalize the spatial and temporal representation of our dataset, we introduce the following definitions.

**Definition 1 (Spatial Graph).** The hydrological observation network is represented as a graph $\mathcal{G} = (\mathcal{V}, \mathcal{E})$, where
$\mathcal{V} = \{v_1, v_2, \ldots, v_N\}$ is the set of $N$ nodes (stations), and $\mathcal{E} \subseteq \mathcal{V} \times \mathcal{V}$ denotes the set of spatial connections (edges). These

edges are encoded using an adjacency matrix $A \in R^{N \times N}$, where $A_{ij}$ denotes the strength or existence of a connection between

node $v_i$ to node $v_j$. The matrix $A$ may be binary or real-valued, directed or undirected, and static or dynamically learned.

**Definition 2 (Feature Matrix).** At each time step $t$, each node $v_i \in \mathcal{V}$ is associated with a feature vector $x_i^t \in R^F$, where

$F = 3$ represents three hydrometeorological variables: streamflow, precipitation and temperature. The complete input
sequence is arranged into a 3D tensor $X \in R^{T \times N \times F}$, where $T$ is the number of time steps.

**Definition 3 (Spatial-Temporal Graph).** A spatiotemporal graph is a dynamic representation that captures both spatial and
temporal dependencies in the hydrological system. It is defined as a sequence of attributed graphs $\mathcal{G}_t = (A_t, X_t)$, where

$A_t \in R^{N \times N}$ is a fixed or learnable adjacency matrix at time step $t$ and $X_t \in R^{N \times F}$ is the feature matrix.

Based on this unified formulation, we consider and compare three strategies for constructing the adjacency matrix $A$:

**(a)   Flow Direction Graph:**

The flow direction graph defines the adjacency matrix $A \in R^{N \times N}$ based on known hydrological flow paths between stations.
It constructs a directed and binary graph by encoding upstream-to-downstream relationships as:

$$A_{ij} = \begin{cases} 1, & \text{if station } v_i \text{ is upstream of station } v_j \\ 0, & \text{otherwise} \end{cases} \tag{1}$$

where $A_{ij}$ indicates a directed edge from station $v_i$ to station $v_j$.

**(b)   Dynamic Time Warping (DTW) Graph**:



In this strategy, the adjacency matrix is constructed from DTW distances between multivariate time-series recorded at each station. It yields a symmetric weighted graph. Given the feature matrix $X \in R^{T \times N \times F}$, where each node has three input features (streamflow, precipitation and temperature), we denote $x_i^{(f)} \in R^T$ as the time series of feature $f$ at station $i$. For each pair of nodes $(i, j)$, we compute DTW distances:

$$d_{ij}^{(f)} = \text{DTW}\left(x_i^{(f)}, x_j^{(f)}\right), \quad f \in \{\text{flow}, \text{prcp}, \text{temp}\} \tag{2}$$

The final adjacency value $A_{ij}$ is computed as a weighted combination of similarity scores derived from DTW distances:

$$A_{ij} = \alpha \exp\left(-\frac{d_{ij}^{\text{prcp}}}{\sigma_p}\right) + \beta \exp\left(-\frac{d_{ij}^{\text{temp}}}{\sigma_t}\right) + \gamma \exp\left(-\frac{d_{ij}^{\text{flow}}}{\sigma_f}\right) \tag{3}$$

where $\sigma_p, \sigma_t, \sigma_f$ are the average DTW distances for each feature across all node pairs, used for normalization. The coefficients $\alpha, \beta, \gamma$ are manually assigned weights satisfying $\alpha + \beta + \gamma = 1$ ($\alpha = 0.4, \beta = 0.3, \gamma = 0.3$) to control the

contribution of each variable.

**(c) Adaptive Graph**:

An adaptive graph avoids using a predefined topology and instead learns its adjacency matrix jointly with the forecasting task. The underlying learning mechanism is detailed in Section 3.2.

Based on the spatio-temporal graph formulation $\mathcal{G}_t = (A, X_t)$, the objective of streamflow forecasting is to learn a mapping

function that captures both spatial and temporal dependencies. Let $Y_{t+1:t+H} = \{Y_{t+1}, \ldots, Y_{t+H}\} \in R^{H \times N}$ denote the streamflow values to be predicted over a future horizon of $H$ steps, where each $Y_{t+h} \in R^N$ represents the runoff at all stations at time $t + h$. The forecasting task is thus formulated as:

$$\hat{Y}_{t+1:t+H} = F_\theta\left(X_{t-T+1:t}, A\right) \tag{4}$$

where $F_\theta$ is a parameterized model that captures both spatial and temporal dependencies for streamflow forecasting.

**3 Methodology**

This section introduces the theoretical method used in this study.

**3.1 CEEMDAN algorithm**

Empirical mode decomposition (EMD) adaptively decomposes a non-stationary signal into a finite set of IMFs based on the signal's intrinsic time scales (Huang et al., 1998). Nevertheless, EMD often suffers from mode mixing, where oscillations

of similar frequency appear in several modes or oscillations of disparate amplitudes are combined in a single IMF. Ensemble EMD (EEMD) mitigates this by repeatedly adding Gaussian white noise and averaging the decompositions (Wu and Huang,



2009), but residual noise can still contaminate the reconstructed signal. CEEMDAN further addresses these shortcomings by guaranteeing completeness and adaptively cancelling the injected noise during reconstruction, thus yielding lower reconstruction error and virtually eliminating mode mixing (Lin et al., 2020). In contrast to wavelet- or Fourier-based

approaches that rely on user-specified basis functions, CEEMDAN is entirely data-driven and adaptive. It decomposes a time series into a finite set of IMFs and a residual component. Each IMF captures oscillatory behavior at different time scales, allowing complex temporal patterns to be analyzed in a more structured manner (Guo et al., 2023). In this study, we employ CEEMDAN to capture the multi-scale characteristics hidden in the hydrological time series. The decomposition yields a collection of IMFs, each capturing variability at a specific temporal scale, along with a residual trend component. Partitioning

the series into scale-specific, quasi-stationary constituents allows the subsequent graph neural network submodels to better capture short-term, medium-term, and long-term streamflow dynamics.

Formally, given a univariate time series $X(t)$, CEEMDAN expresses it as a sum of $K$ intrinsic mode functions and a residual term:

$$X(t) = \sum_{k=1}^{K} \text{IMF}_k(t) + r_K(t) \tag{5}$$

where $\text{IMF}_k(t)$ denotes the $k$-th intrinsic mode function capturing components from high to low frequency, $r_K(t)$ is the final residual after extracting $K$ IMFs.

### 3.2 Adaptive graph learning

A fundamental challenge in applying GNNs to hydrological forecasting is accurately and flexibly defining spatial dependencies among hydrological stations. Static graph structures, though widely adopted, are limited by their basin-specific and time-

invariant properties, which restrict their ability to generalize across regions or adapt to evolving hydrological dynamics. Inspired by GraphWaveNet (Wu et al., 2019), we learn a directed, row-normalized adjacency matrix directly from the data:

$$A_{adp} = \text{softmax}_{\text{row}}\left(\text{ReLU}\left(E_1 E_2^\top\right)\right) \tag{6}$$

where $E_1, E_2 \in R^{N \times e}$ are two trainable node-embedding matrices, $N$ is the number of stations and $e$ the embedding dimension.

This formulation enables the learning of directed spatial dependencies directly from the data.





## 3.3 Spatial graph gated recurrent unit (SGGRU) module

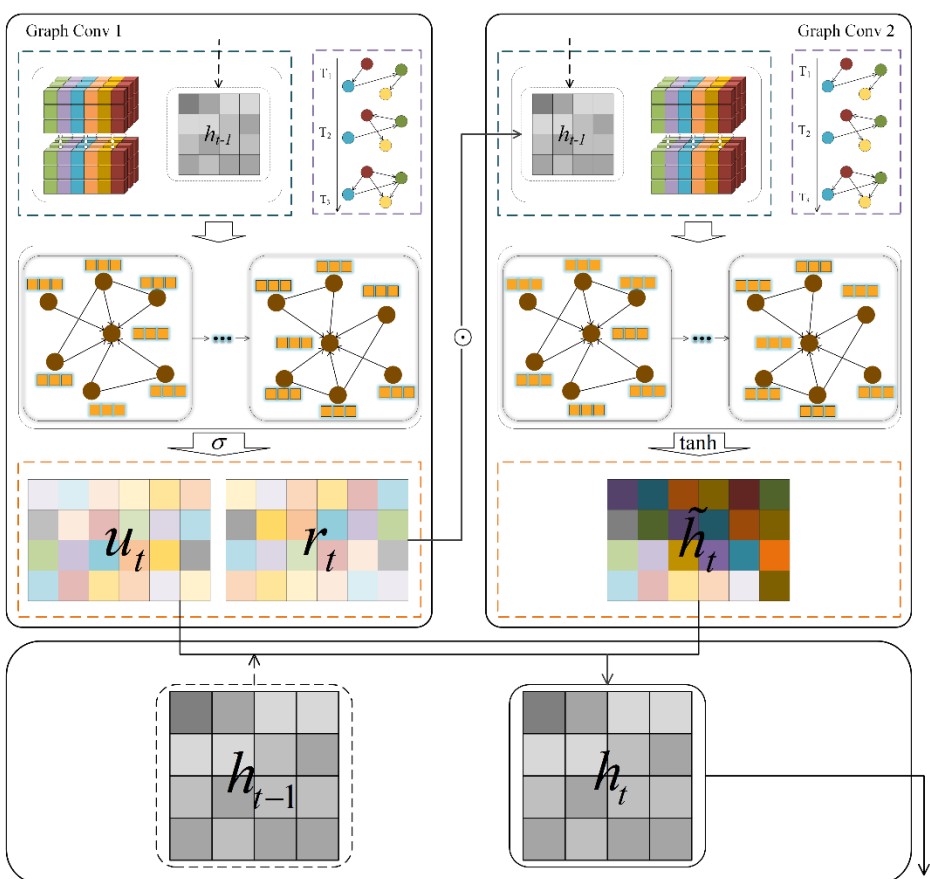

**Figure 2: Architecture of the SGGRU module.**

Following the standard GRU, we extend its capabilities by integrating spatial dependencies via graph convolution. Specifically, we design a spatial graph gated recurrent unit (SGGRU) module, which enhances the recurrent unit's ability to model spatiotemporal dynamics across hydrological stations (Zhao et al., 2020). As illustrated in Fig. 2, SGGRU architecture incorporates graph convolution operations into both the update and reset gate computations, as well as the candidate hidden state generation. This structure allows the model to simultaneously capture spatial correlations among stations (via a learned

or predefined adjacency matrix) and temporal dependencies within the time series.

Mathematically, the SGGRU can be expressed as follows:

$$\left[r_t, u_t\right] = \sigma\left(\text{GraphConv1}\left(\left[x_t, h_{t-1}\right], A_{adp}\right)\right) \tag{7}$$

$$\tilde{h}_t = \tanh\left(\text{GraphConv2}\left(\left[x_t, r_t \odot h_{t-1}\right], A_{adp}\right)\right) \tag{8}$$





$$h_t = u_t \odot h_{t-1} + (1 - u_t) \odot \tilde{h}_t \qquad (9)$$

where $u_t$, $r_t$ and $\tilde{h}_t$ are the update, reset, and potential cell state at time t, respectively. $h_{t-1}$ is the output of the hidden

layer at time $t-1$. $x_t$ is the input at time t. GraphConv1($\cdot$) and GraphConv2($\cdot$) are the graph convolution layers

parameterized by learnable weights. σ is the sigmoid function. tanh is the hyperbolic tangent, and $\odot$ is the element-

wise multiplication.

This architecture ensures that each recurrent update accounts for both the temporal evolution of streamflow and

meteorological features, as well as the spatial dependencies among stations. It accommodates either fixed or learned adjacency

matrices. In this study, all graph-based models, including DTWSGGRU, FDSGGRU, ASGGRU, and their CEEMDAN-

enhanced variants share a common SGGRU backbone to ensure architectural parity across spatial representations. We use the

Adam optimizer for training and other training settings are discussed in Section 4.

### 3.4 Proposed LCEEMDAN-ASGGRU model

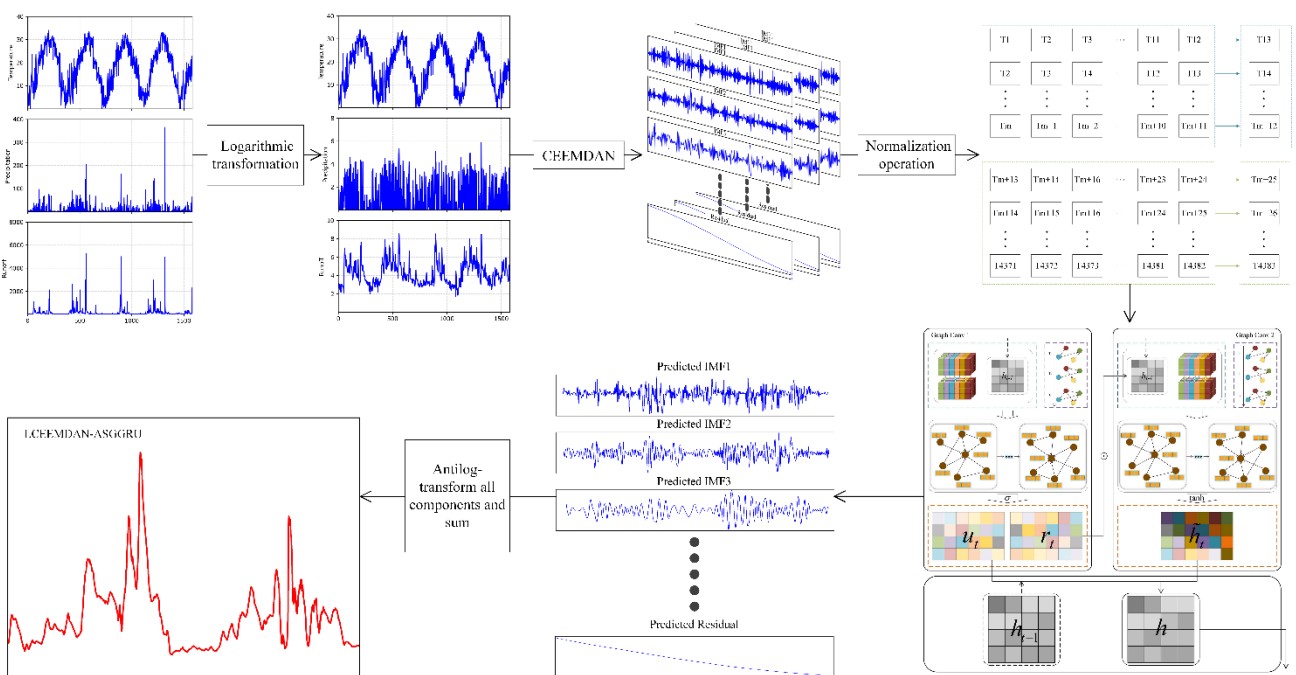


**Figure 3: Framework of LCEEMDAN-ASGGRU.**

To exploit the strengths of multiscale decomposition and adaptive spatial-temporal modeling, we propose a hybrid

LCEEMDAN-ASGGRU. The specific modeling process is shown in Fig. 3.

(1)   To stabilize variance and mitigate the influence of extreme values observed in streamflow and precipitation time series,





we apply a logarithmic transformation prior to decomposition. Specifically, we adopt the natural logarithm with the transformation defined as:

$$X_{\log}(t) = \log(1 + X(t)) \tag{10}$$

This transformation ensures numerical stability while reducing the influence of large outliers, thereby enhancing the subsequent CEEMDAN decomposition and improving the separability of IMFs across scales.

(2) The CEEMDAN decomposition was performed using the PyEMD library with default parameters, including a noise standard deviation of 0.2 and 250 noise-assisted trials. Each variable (streamflow, precipitation, and temperature) at each station was decomposed into eight IMFs and one residual component.

(3) Each IMF and residual component is standardized using z-score normalization before forecasting to ensure they remain on the same scale:

$$X'(t) = \frac{X_{\log}(t) - \bar{X}_{\log}(t)}{\sigma} \tag{11}$$

where $\bar{X}_{\log}(t)$ denote the mean deviation of the series and $\sigma$ denote the mean and standard deviation of the series. The corresponding normalization parameters were stored and applied during postprocessing to enable accurate inverse transformation of the model predictions back to the original scale.

(4) Each of the nine decomposed components from the CEEMDAN process is treated as an independent prediction subtask. For each component, a separate instance of the ASGGRU model is trained independently, allowing the model to specialize in capturing the temporal dynamics unique to that frequency scale. Notably, each IMF and the residual has its own set of optimal hyperparameters, reflecting the varying statistical characteristics and predictive complexities across components. This design provides additional flexibility, enabling the model to allocate capacity appropriately.

(5) During inference, each trained submodel outputs a predicted sequence corresponding to its specific component. These outputs are then linearly aggregated across all components to reconstruct the final streamflow prediction. Mathematically, the reconstruction can be expressed as:

$$\hat{y}(t) = \sum_{k=1}^{K} \hat{y}_k(t) \tag{12}$$

where $\hat{y}_k(t)$ denotes the prediction from the IMF or residual submodel, $\hat{y}(t)$ is the final reconstructed streamflow prediction at time step $t$.

The proposed hybrid LCEEMDAN-ASGGRU model integrates a logarithmic transformation for variance stabilization, the CEEMDAN technique for multiscale decomposition, and an ASGGRU for spatial-temporal modeling. This integrated framework enables the model to effectively capture nonlinear, multiscale hydrometeorological dynamics and complex inter-station dependencies. For comparison, the CEEMDAN-ASGGRU model employs the same framework as the LCEEMDAN-ASGGRU but excludes the logarithmic transformation.





### 3.5 Evaluation indicators

To evaluate the forecast accuracy of streamflow prediction models, two widely adopted performance metrics are used: root mean squared error (RMSE) and the Nash–Sutcliffe efficiency coefficient (NSE). Together, these two metrics provide a balanced assessment of both the error magnitude and the explained variance in the predictions. RMSE and NSE are calculated as follows:

$$RMSE = \sqrt{\frac{1}{N}\sum_{i=1}^{N}(y_i - \hat{y}_i)^2} \tag{13}$$

$$NSE = 1 - \frac{\sum_{i=1}^{N}(y_i - \hat{y}_i)^2}{\sum_{i=1}^{N}(y_i - \bar{y})^2} \tag{14}$$

### 3.6 Forecasting uncertainty

HMM-GMR (Calinon and Billard, 2007) is employed as a post-processing method to quantify the predictive uncertainty of the trained models. This approach enables estimation of the full conditional distribution of the observed streamflow, given the model's point forecast, and is capable of handling heteroscedastic and non-normal error structures, which are common in hydrological time series (Chen et al., 2016).

In the proposed framework, we first construct a joint sequence consisting of the predicted and observed streamflow values, $X = \{(x_t^{\mathrm{pred}}, x_t^{\mathrm{obs}})\}_{t=1}^{T}$ for each model at each target station. An HMM with $K$ hidden states is trained on this bivariate sequence using the expectation-maximization (EM) algorithm. In our implementation, the number of hidden states is fixed at $K = 3$, balancing modeling flexibility and stability while maintaining low computational complexity. Each hidden state corresponds to a bivariate Gaussian distribution, and the hidden state sequence evolves according to a first-order Markov chain.

Let $\mu_k = \left[\mu_1^{(k)}, \mu_2^{(k)}\right]$ and $\Sigma_k = \begin{bmatrix} \Sigma_{11}^{(k)} & \Sigma_{12}^{(k)} \\ \Sigma_{21}^{(k)} & \Sigma_{22}^{(k)} \end{bmatrix}$ be the mean vector and covariance matrix of the bivariate distribution in state $k$. The conditional distribution of the observation given the prediction is then derived using Gaussian Mixture Regression (GMR). The final predictive distribution is a weighted combination of state-wise conditional Gaussians:

$$p(x_t^{\mathrm{obs}} \mid x_t^{\mathrm{pred}}) = \sum_{k=1}^{K} w_k(x_t^{\mathrm{pred}}) \cdot \mathcal{N}(x_t^{\mathrm{obs}}; \mu_{2|1}^{(k)}, \Sigma_{2|1}^{(k)}) \tag{15}$$

where $\mu_{2|1}^{(k)} = \mu_2^{(k)} + \Sigma_{21}^{(k)}\left(\Sigma_{11}^{(k)}\right)^{-1}\left(x_t^{\mathrm{pred}} - \mu_1^{(k)}\right)$, $\Sigma_{2|1}^{(k)} = \Sigma_{22}^{(k)} - \Sigma_{21}^{(k)}\left(\Sigma_{11}^{(k)}\right)^{-1}\Sigma_{12}^{(k)}$ and the weights $W_k$ are computed via the forward algorithm of the HMM.

Using the resulting predictive distribution at each time step, we evaluate the uncertainty performance of each model based on three standard metrics:





(1) Interval Coverage Probability (ICP)

The ICP measures the proportion of observed values falling within the model's prediction interval at a given confidence level $\alpha$. A well-calibrated model is expected to produce an ICP close to the nominal confidence level.

$$\text{ICP} = \frac{1}{T}\sum_{t=1}^{T}\mathbb{I}\left(x_t^{\text{obs}} \in \left[\mu_{2|1}^{(k_t)} - z_\alpha \cdot \sigma_{2|1}^{(k_t)}, \mu_{2|1}^{(k_t)} + z_\alpha \cdot \sigma_{2|1}^{(k_t)}\right]\right) \tag{16}$$

where $\mathbb{I}$ is the indicator function, $Z_\alpha$ is the standard normal quantile, and $\mu_{2|1}^{(k_t)}, \sigma_{2|1}^{(k_t)}$ are the predictive mean and standard

deviation estimated from the HMM-GMR at time $t$.

(2) Prediction Interval Width (PIW)

The PIW quantifies the average width of the prediction interval at a given confidence level. It reflects the sharpness of the predictive distribution: narrower intervals are preferred when coverage is adequate.

$$\text{PIW} = \frac{1}{T}\sum_{t=1}^{T}\left(2 \cdot z_\alpha \cdot \sigma_{2|1}^{(k_t)}\right) \tag{17}$$

(3) Continuous Ranked Probability Score (CRPS)

CRPS evaluates the quality of the full predictive distribution by measuring the squared difference between the predicted cumulative distribution function $F_t(\cdot)$ and the empirical step function at the observation:

$$\text{CRPS} = \frac{1}{T}\sum_{t=1}^{T}\int_{-\infty}^{+\infty}\left[F_t(x) - \mathbb{H}(x - x_t^{\text{obs}})\right]^2 dx \tag{18}$$

where $\mathbb{H}(\cdot)$ is the Heaviside step function.

## 4 Results and discussion

### 4.1 Experimental configuration

To ensure a fair and consistent evaluation across all models, we adopt a unified experimental configuration. The feature matrix and target time series are split into training and testing subsets using an 8:2 ratio. Each model receives the same multivariate

input tensor $X_{t-11:t} \in R^{12 \times N \times 3}$ (12-day history of streamflow, precipitation and temperature at the $N = 14$ stations) and

predicts next-day streamflow $\hat{Y}_{t+1} \in R^N$.

### 4.2 Model settings

#### 4.2.1 Baseline and Graph-based models

We construct four deep learning models: LSTM (as a non-graph baseline model), and three GNN-based models—

DTWSGGRU, FDSGGRU, and ASGGRU. All graph models are implemented using the SGGRU introduced in Section 3.3.





For consistency and fair comparison, all models share the same training settings: a batch size of 32, a learning rate of 0.001 and 1000 training epochs. SGGRU module adopts a two-layer graph convolution structure with a hidden dimension of 64, and it is optimized using Adam. Training is conducted with the Huber loss function, which enhances robustness to outliers and abrupt fluctuations commonly present in streamflow data. Model performance is evaluated using two widely adopted metrics,

RMSE and NSE (Section 3.5). Each experiment is repeated 10 times with different random seeds to ensure reliable results.

(1) **LSTM:** The LSTM network serves as a classical baseline for sequence modeling. It consists of three stacked LSTM layers.

(2) **DTWSGGRU:** This model uses a fixed spatial graph constructed based on DTW distances computed from the historical dataset between pairs of stations. This graph captures temporal similarity patterns across stations.

(3) **FDSGGRU:** This model utilizes a binary (0-1) flow-direction adjacency matrix representing the hydrological network, where edges denote directed upstream–downstream flow relationships. This structure encodes physical connectivity within the river system.

(4) **ASGGRU:** ASGGRU integrates a data-driven adaptive graph into the SGGRU module. Instead of using a predefined static graph, the model dynamically learns spatial dependencies during training through a node embedding mechanism. The

embedding dimension is set to 16.

### 4.2.2 Decomposition-based variants

For decomposition-based approaches, the original dataset is first decomposed into IMFs and a residual. Each decomposition component is then trained using a separate ASGGRU model, which is optimized independently. For every submodel, we conduct a 5-fold cross-validation to explore the hyperparameter space (Table 1), including hidden and embedding dimensions,

batch size, learning rate, weight decay, and maximum epochs. The CEEMDAN-ASGGRU model uses the same submodel architecture and hyperparameter configurations as the LCEEMDAN-ASGGRU model.

### 4.3 Prediction performance comparison

To validate the effectiveness of the proposed LCEEMDAN-ASGGRU, this section presents a comprehensive performance analysis through comparative experiments. Instead of evaluating each model in isolation, we adopt an ablation approach,

progressively breaking down the proposed model into its key components: the adaptive graph learning module, the multi-scale CEEMDAN decomposition, and the logarithmic transformation. This hierarchical evaluation allows us to isolate and quantify the contributions of the spatial and temporal enhancements embedded in the architecture.




### 4.3.1 Overall accuracy

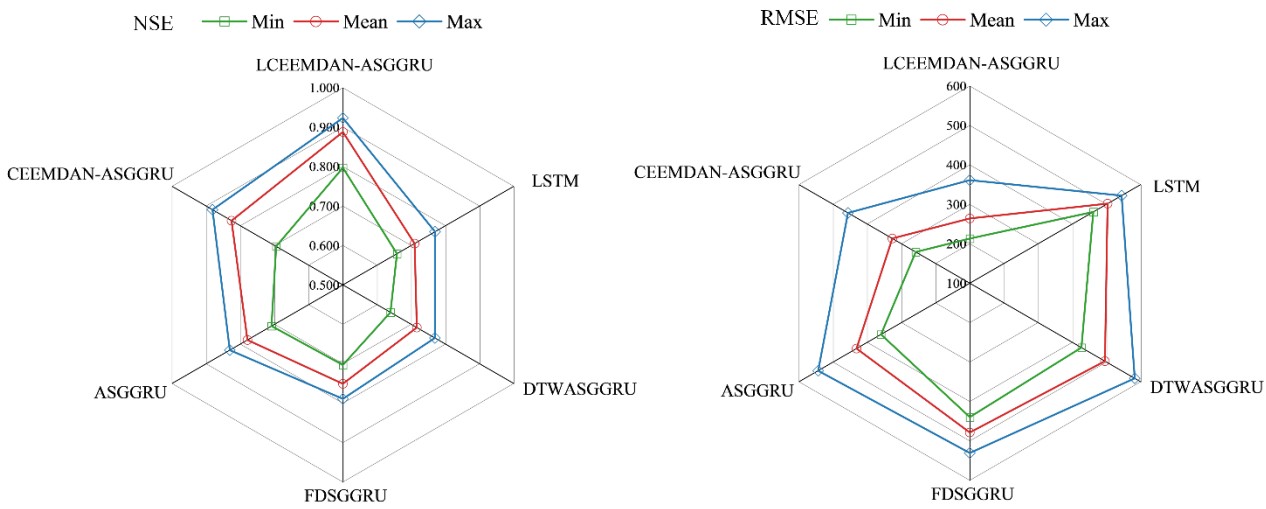

**Figure 4: Model performance comparison across different forecasting models.**

The overall forecasting performance of the six models is summarized in Fig. 4. The DTWSGGRU, FDSGGRU, and ASGGRU all outperform the non-graph LSTM baseline, indicating that incorporating spatial dependencies through graph-based architectures enhances predictive skill. This observation is consistent with the findings of Yang et al. (2023), which demonstrated that integrating spatial graph structures significantly enhances runoff prediction accuracy compared to non-graph baselines in multi-station experiments. The ASGGRU achieves higher accuracy than the two static graph models, highlighting the benefits of adaptively learning spatial relationships from data rather than relying solely on predefined graph structures.

In addition, the integration of multiscale decomposition through CEEMDAN substantially enhances model performance, and the application of logarithmic transformation prior to decomposition yields the best overall accuracy. The LCEEMDAN-ASGGRU model achieves the lowest mean RMSE of 264 and the highest mean NSE of 0.888, representing the best overall predictive performance.

These overall findings provide a consistent foundation for the following analyses, where the specific contributions of graph construction and decomposition strategies are examined in greater detail at both the model level and the station level.





### 4.3.2 Effect of graph construction

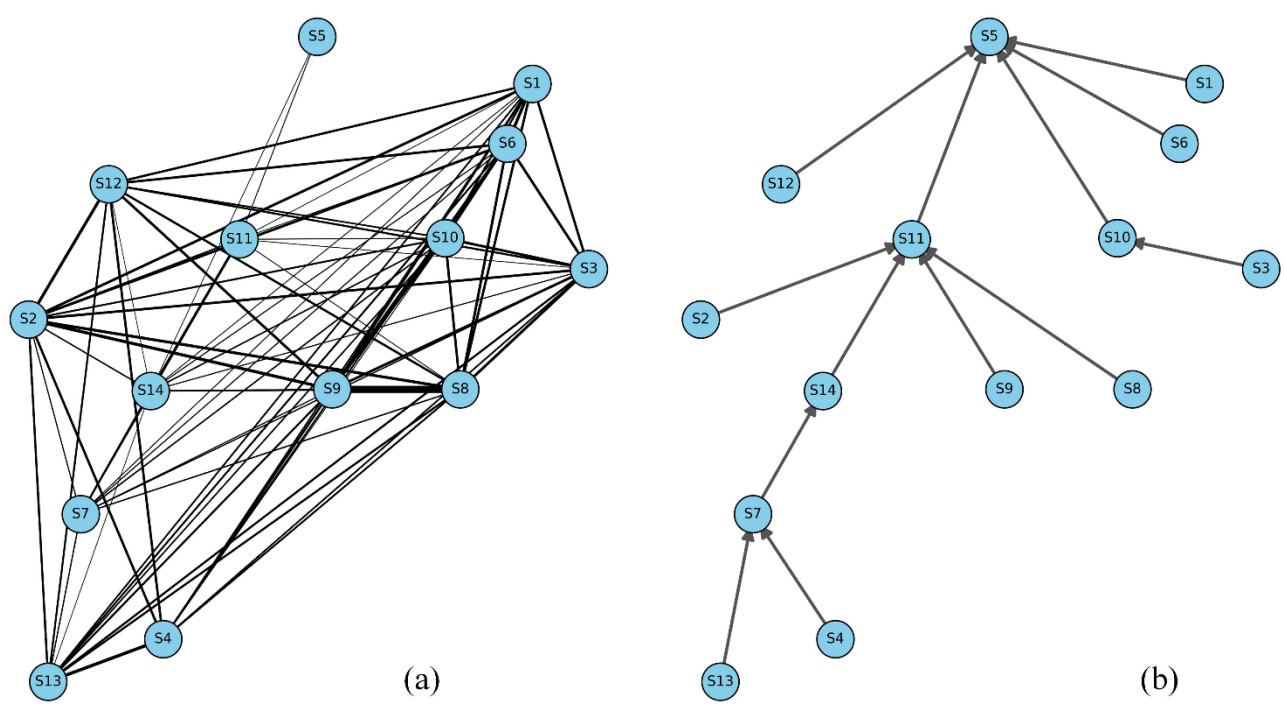


**Figure 5: Two statics graph, (a) DTW-similarity graph, an undirected, weighted graph, edge width is proportional to similarity, and only connections above 0.3 threshold are shown; (b) Flow-direction graph, a directed graph based on the river-flow adjacency matrix, where arrows indicate upstream-to-downstream connectivity.**

To examine the impact of different spatial graph structures on streamflow prediction, we compared three variants of the proposed architecture that differed only in how their spatial adjacency matrices were constructed: DTWSGGRU, FDSGGRU, and ASGGRU. The DTWSGGRU employs a static similarity graph whose edge weights are given by DTW distances between historical series, thereby capturing temporal synchrony among stations (Fig. 5 (a)). The FDSGGRU model adopts a binary flow-direction graph to encode hydrological connectivity based on river topology (Fig. 5 (b)). Both graphs are predefined and

fixed throughout training. In contrast, the ASGGRU model learns an adaptive spatial graph via node embeddings that are updated jointly with the model parameters.

As shown in Fig. 4, the overall RMSE and NSE scores reveal that all three GNN models outperform the LSTM baseline, highlighting the benefit of incorporating spatial structure. Among the static graph models, FDSGGRU outperforms DTWSGGRU on average (mean NSE: 0.751 vs. 0.716; mean RMSE: 479 vs. 496), suggesting that when a GNN adopts the

flow-direction adjacency matrix, its spatial representation aligns more closely with the basin's hydraulic connectivity, leading to improved streamflow prediction accuracy. This is consistent with Sun et al. (2022), which found that incorporating flow direction in spatial graphs leads to more physically consistent hydrological predictions across diverse basins.



However, as shown in Tables 2 and 3, neither DTWSGGRU nor FDSGGRU consistently outperforms LSTM at every station. For example, DTWSGGRU underperforms LSTM at stations S1, S4, S5, and S13, and FDSGGRU lags behind LSTM at
station S5 in both NSE and RMSE. These discrepancies illustrate the limitations of static graphs: DTWSGGRU is constrained by non-physical similarity measures, while FDSGGRU, though physically grounded, uses fixed edge weights that fail to reflect varying connection strengths across different hydrological regimes. This limitation has also been emphasized in recent graph learning literature. A recent survey on dynamic graph neural networks pointed out that static graphs "limit the ability to model complex and time-varying spatial relationships" (Zheng et al., 2024), reinforcing the necessity of adaptive graph construction
in systems with evolving spatial dependencies.

In contrast, ASGGRU achieves higher mean NSE and lower mean RMSE than LSTM at all stations. Its ability to dynamically learn spatial correlations allows it to capture both physically plausible structures (as in FDSGGRU) and temporal alignment patterns (as in DTWSGGRU). This flexibility is important in basins with complex hydrological processes or variable forcing-response dynamics. These findings suggest that while predefined graphs offer modest benefits over non-graph baselines,
adaptive graph learning provides the most consistent and substantial improvements, validating its importance as a core graph learning component of the proposed framework.

### 4.3.3 Improvement of decomposition and logarithmic transformation

In this subsection, we investigated how streamflow forecasting could be further enhanced by introducing multiscale temporal decomposition and logarithmic transformation, focusing on two augmented models: CEEMDAN-ASGGRU and
LCEEMDAN-ASGGRU.

The CEEMDAN-ASGGRU model extended ASGGRU by incorporating CEEMDAN decomposition into the modeling pipeline. As delineated in Section 3.4, the streamflow, precipitation, and temperature time series at each station were decomposed into eight IMFs and one residual component. Each of these components was modeled independently using a dedicated ASGGRU model. This enableed the architecture to explicitly learn temporal dynamics at varying frequency scales.
From Fig. 4, CEEMDAN-ASGGRU demonstrated a 24.27% reduction in mean RMSE and a 6% increase in mean NSE compared to ASGGRU. Such performance improvements were consistent with prior hydrological modeling. Xu et al. (2024) reported that a hybrid CEEMDAN-based model reduced RMSE by 60-70% in monthly runoff prediction for two stations in China.

The LCEEMDAN-ASGGRU employed a log-transformation of the time series prior to CEEMDAN decomposition. The
objective of this step was to stabilize the variance and mitigate the impact of extreme streamflow values. The LCEEMDAN-ASGGRU model exhibited the lowest mean RMSE of 264 and the highest mean NSE of 0.888 (Fig. 4), demonstrating superior performance in comparison to all the other models evaluated in this study. Six stations (S5, S7, S10, S11, S13, S14) achieved NSE values above 0.900, while station S1 records the largest improvement over the LSTM ($\Delta$NSE = +0.260) (Table 3).

To further visualize these improvements, Figure 6 presents the observed and predicted streamflow of six models at two
representative stations: S4, located in the southernmost part of the basin and exhibiting the lowest mean streamflow, and S5,





located in the northernmost end and recording the highest mean streamflow. In addition, a partially enlarged subgraph is provided for clearer observation. From Fig. 6, we found that the forecast results of the LCEEMDAN-ASGGRU model were very consistent with the observed streamflow. At both hydrological extremes, the proposed model outperformed the five other models in tracking changes in streamflow dynamics.

The step-wise performance gains show that each added module supplies a distinct, non-overlapping piece of the forecasting puzzle. The adaptive graph structure continuously recalibrates edge weights, reflecting the strength of spatial relationships and the degree of similarity between sites. The CEEMDAN decomposition disassembles the original flow sequence into mutually unmixed multi-scale IMFs, thereby enabling the model to learn flow patterns at different scales separately. The logarithmic transformation stabilizes variance and compresses extreme peaks. The combination of these three ingredients produces a

uniformly superior model that achieves accurate prediction of streamflow of varying scales.

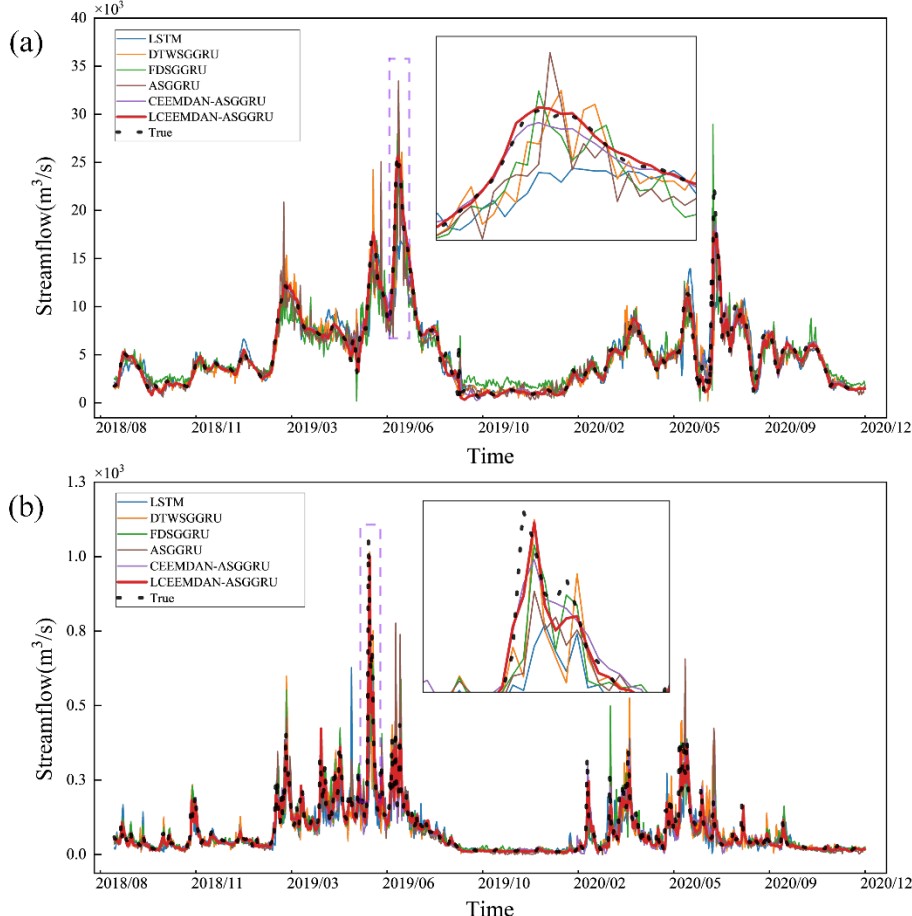

**Figure 6: Observed and predicted streamflow at two stations: (a) S4, located at the southernmost point of the basin and characterized by persistently low-flow conditions; (b) S5, situated at the northernmost point and exhibiting frequent and extreme high-flow events.**





## 4.4 Spatial interpretability analysis

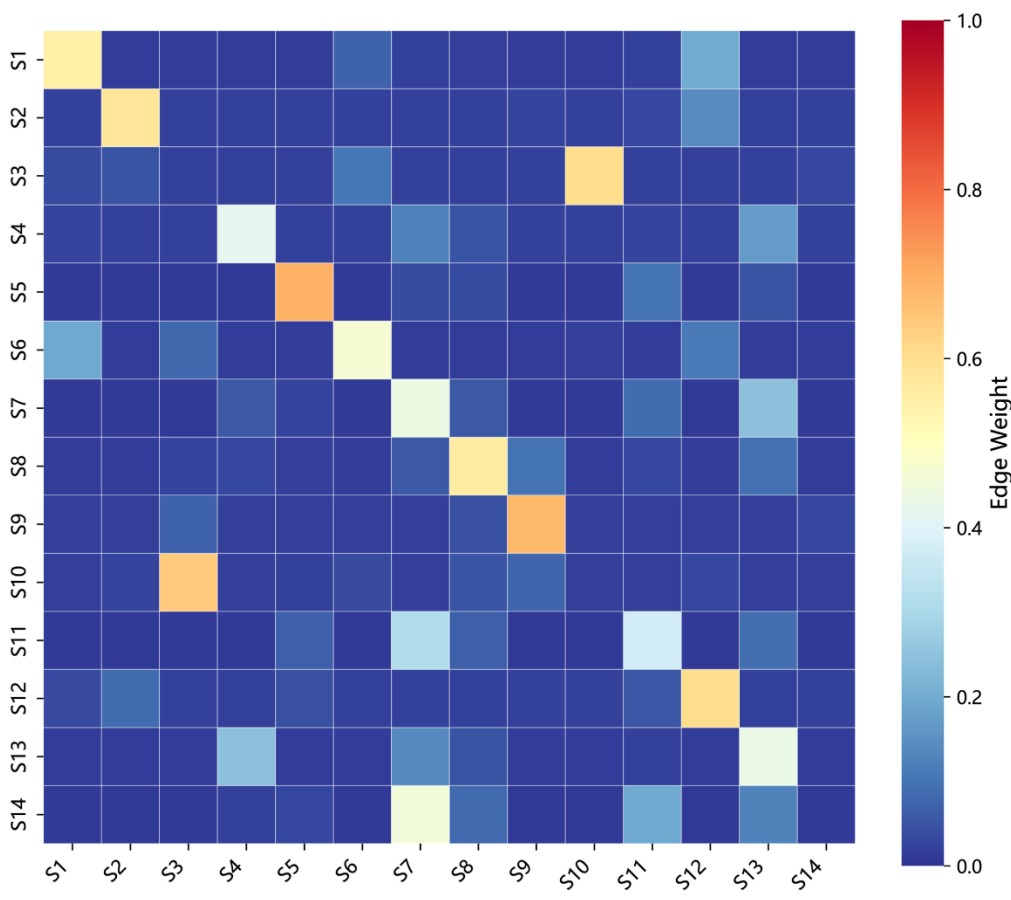


**Figure 7: Learned adaptive adjacency matrix obtained from ASGGRU.**

To investigate the spatial representations captured by the adaptive graph learning mechanism, we analyze the learned graph obtained from ASGGRU and evaluate its hydrological consistency, its relationship with static graph priors, and its overall

interpretability.

The learned adaptive adjacency matrix is visualized in Fig. 7. Several strong directional connections emerge from the learned structure, many of which correspond well with known flow-direction relationships within the river network. In particular, connections such as S3→S10, S4→S14, S7→S11, and S13→S7 reflect physically consistent upstream–downstream dependencies that are embedded in the true hydrological topology. Hence, the data-driven learning process extracts physically

meaningful connectivity without any explicit spatial supervision. This result echoes findings by Bai and Tahmasebi. (2023), who have also demonstrated that learned adjacency matrices from adaptive GNNs can reflect meaningful spatial structures even without explicit supervision in groundwater forecasting and environmental modeling. Besides these hydrologically





plausible edges, the matrix displays indirect or cross-basin connections that may capture remote influences or shared meteorological forcing.

To further quantify the relationship between the learned adaptive graph and the two static priors, we conducted an overlap analysis under varying edge-weight thresholds (0.05, 0.10, and 0.20). Table 4 summarizes the overlap analysis results between the learned adaptive adjacency matrix and the predefined FD and DTW graphs under different thresholds, which provides quantitative insights into the extent of consistency and divergence between data-driven and prior-based graph structures. The adaptive graph demonstrates strong consistency with the flow-direction prior. At the 0.05 threshold, 80.5% of the A edges

overlap with FD edges, indicating that the adaptive graph learning model effectively captures physically plausible flow connectivity patterns. In contrast, only 32.1% of A edges coincide with DTW-based similarity edges, reflecting the model's more selective incorporation of temporal similarity information.

This overlap pattern remains stable as the threshold increases. As the edge-weight threshold rises from 0.05 to 0.20, the total number of effective A edges decreases from 72 to 45. Across the examined thresholds, the FD-A overlap consistently exceeds

70%, while the DTW-A overlap remains below 20%. Importantly, across all thresholds, no A-only edges were identified, indicating that the adaptive graph learning primarily operates within the combined subspace spanned by flow-direction and temporal similarity priors, rather than introducing entirely novel spatial connections. This behavior suggests that the adaptive graph learning model effectively refines and re-weights hydrologically and temporally meaningful spatial relationships, balancing physical consistency with data-driven flexibility.



**4.5 Uncertainty estimation**

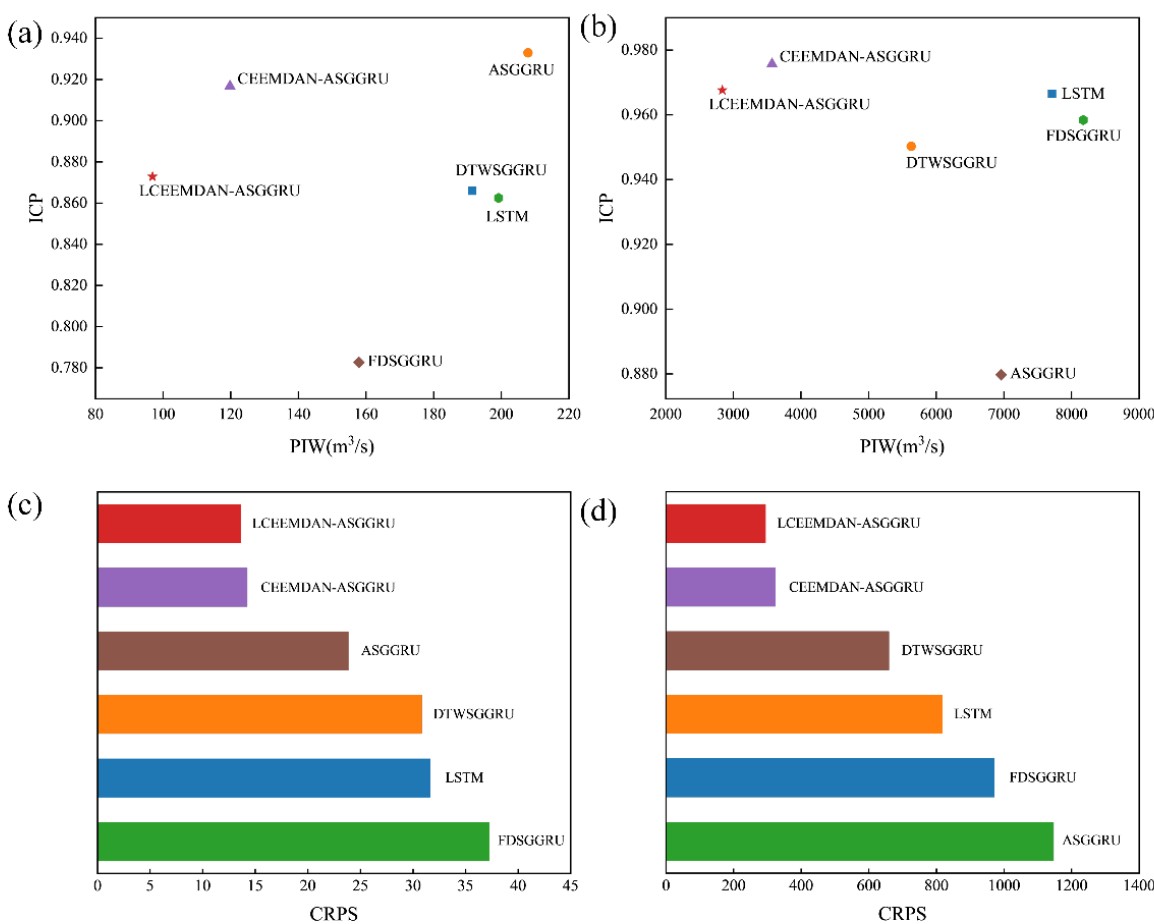

**Figure 8: Predictive uncertainty performance of six models evaluated using three metrics: ICP, PIW, and CRPS. (a) and (c) show the results for S4, while (b) and (d) correspond to S5.**

In this subsection, HMM-GMR is employed to evaluate the probabilistic reliability of each model output. Figure 8 presents the joint evaluation of PIW, ICP, and CRPS at two stations, S4 and S5. PIW measures the sharpness of prediction intervals, ICP evaluates their calibration, and CRPS provides a comprehensive score that jointly reflects both calibration and sharpness. Ideally, a well-performing model should maintain a narrow PIW, achieve an ICP close to the nominal 95% confidence level, and minimize the CRPS.

For station S4, LCEEMDAN-ASGGRU achieves the narrowest intervals (PIW = 96.9 m³/s) with an ICP of approximately 0.880, indicating a favorable balance between precision and reliability. Meanwhile, CEEMDAN-ASGGRU attains a slightly



higher ICP (0.920), but at the cost of noticeably wider intervals. LSTM and DTWSGGRU produce broad intervals (PIW > 190 m³/s) but still fall short of nominal coverage, suggesting both overdispersion and underconfidence in their uncertainty estimation. ASGGRU yields a favorable ICP (0.930) but also the widest intervals among graph-based models. For station S5,

both LCEEMDAN-ASGGRU and CEEMDAN-ASGGRU maintain ICP values above 0.960, with PIWs constrained to a moderate range (3000–4000 m³/s). In contrast, ASGGRU exhibits a sharp decline in ICP (0.880) and has the broadest intervals (PIW > 6500 m³/s).

To provide a more comprehensive evaluation, we further compute the CRPS (Fig. 8), which accounts for both calibration and sharpness across the full predictive distribution. LCEEMDAN-ASGGRU outperforms all other models at both stations,

followed by CEEMDAN-ASGGRU. Although ASGGRU ranks third at station S4, its performance deteriorates sharply at station S5, recording the poorest CRPS among all models. These results indicate that, despite its adaptive spatial graph, ASGGRU fails to adequately distinguish high- from low-frequency flow variability, confirming that spatial adaptivity alone is insufficient for robust generalization under strongly heterogeneous hydrological conditions. By contrast, the decomposition-based variants excel under extreme flow regimes, underscoring the essential role of multiscale decomposition and

demonstrating the additional benefit of logarithmic transformation in stabilizing variance and enhancing predictive skill.

Figure 9 offers a time-series visualization of predicted means and associated 95% confidence intervals for all six models at stations S5 and S4. At both stations, LCEEMDAN-ASGGRU provides the closest alignment between predicted and observed flows while maintaining the narrowest and most adaptive uncertainty bands. In high-flow scenarios (S5), it effectively tracks peak magnitudes without excessive widening of intervals; in low-flow scenarios (S4), it captures subtle variations with both

tight intervals and consistent coverage.

The results of the uncertainty analyses confirm that the proposed model not only achieves the highest deterministic accuracy but also produces the most reliable and informative uncertainty estimates, making it particularly suitable for applications requiring robust risk-aware streamflow forecasting across diverse hydrological conditions.



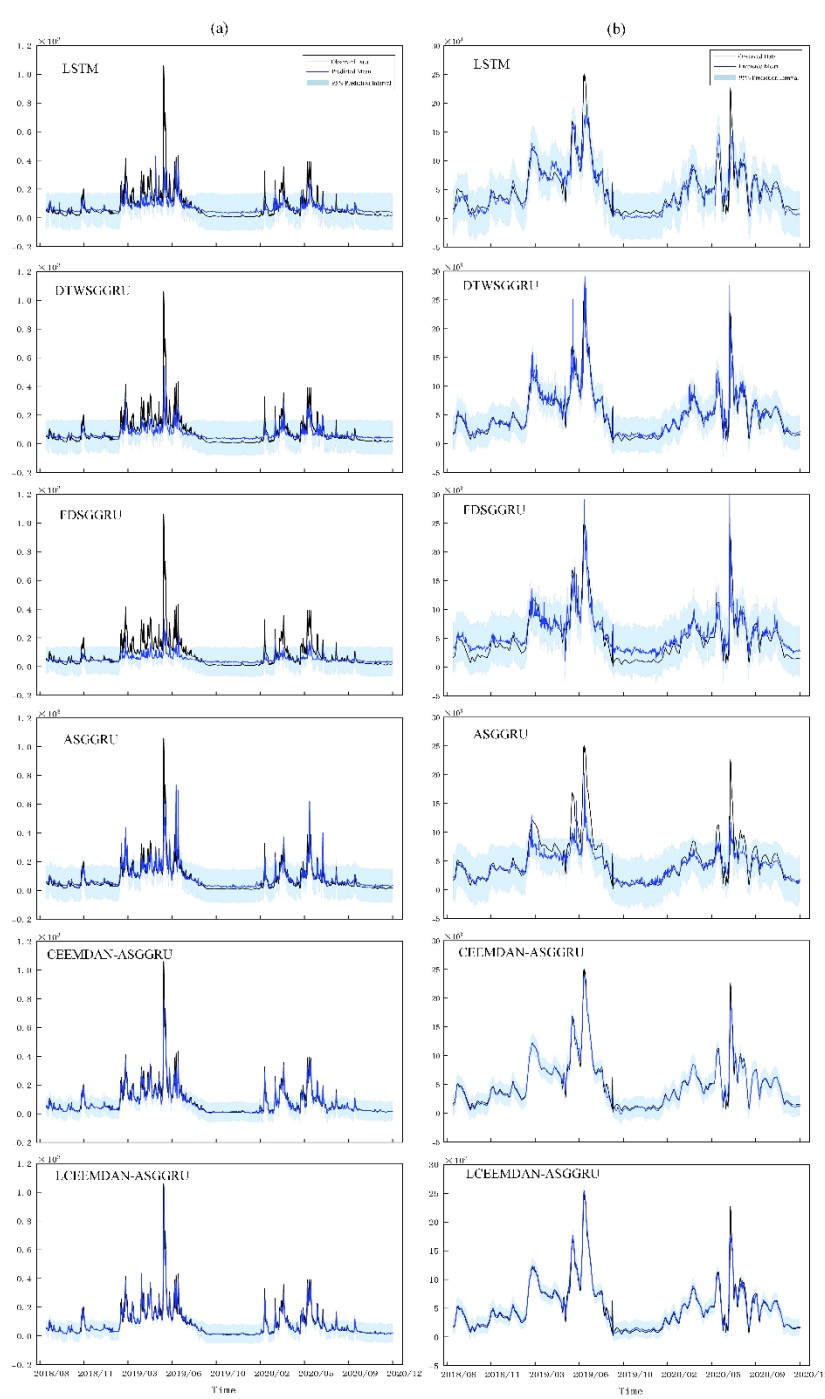

**Figure 9: Comparison of the mean of the six models with the magnitude range of the observed flows: (a) S4, (b)S5.**



## 5. Conclusion

In this study, we propose a hybrid LCEEMDAN-ASGGRU model, which combines logarithmic transformation, multiscale time decomposition, and adaptive graph modeling, for streamflow forecasting. The hybrid LCEEMDAN-ASGGRU is applied in the Poyang Lake basin and compared with the CEEMDAN-ASGGRU, ASGGRU, DTWSGGRU, FDSGGRU, and LSTM. The results show that the hybrid model significantly outperforms the five other models in terms of performance metrics. Specifically, LCEEMDAN-ASGGRU achieves the lowest RMSE and the highest NSE, and is effective in capturing high-flow peaks and low-flow variations. The analysis of the learned graph structure shows that the adaptive graph module effectively refines and re-weights hydrologically and temporally meaningful spatial connections, striking a balance between physical consistency and data-driven flexibility. In addition, HMM-GMR uncertainty analysis confirms that LCEEMDAN-ASGGRU delivers the most stable and reliable probabilistic forecasts, exhibiting the lowest CRPS and comparatively narrow prediction intervals. In summary, the proposed model can produce accurate and reliable prediction results, which can support water resource managers in making decisions regarding resource allocation and reservoir operation.

### Acknowledgements

This work was supported by the National Natural Science Foundation of China (42301041 and 42301533), and Jiangxi Provincial Natural Science Foundation (20252BAC240108).

### Data and code availability statement

The data and relevant codes can be requested through Lizhi Tao.

### Author contributions

Yueming Nan: Conceptualization, Software, Formal analysis, Data Curation , Writing - Original Draft, Visualization. Lizhi Tao: Conceptualization, Supervision, Validation, Writing - Review & Editing. Dong Yang: Conceptualization, Validation, Supervision. Haibo Zou: Methodology, Writing - Review & Editing, Supervision. Zhichao Cui: Visualization, Validation. Yuanbo Luo: Validation, Data Curation, Visualization.

### Competing interests

The authors declare that they have no competing interests.

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



Table 1 Hyperparameter search space for IMF and residual submodels.

| Hyperparameters | Search Range |
| --- | --- |
| Hidden dimension | {16, 32, 64, 128} |
| Embedding dimension | {8, 16, 32} |
| Batch size | {16, 32, 64} |
| Maximum epochs | {500, 800, 1000, 1500, 2000} |
| Learning rate | {0.01, 0.001, 0.005, 0.0001, 0.0005} |
| Weight decay | {0.0, 0.001, 0.0001} |



Table 2 RMSE of the six models at 14 stations (S1–S14), averaged over 10 independent runs.

| Model | RMSE | | | | | | | | | | | | | |
|---|---|---|---|---|---|---|---|---|---|---|---|---|---|---|
| | S1 | S2 | S3 | S4 | S5 | S6 | S7 | S8 | S9 | S10 | S11 | S12 | S13 | S14 |
| LSTM | 312 | 153 | 312 | 58 | 1646 | 297 | 801 | 236 | 137 | 475 | 1152 | 126 | 279 | 1064 |
| DTWSGGRU | 333 | 141 | 300 | 59 | 1735 | 295 | 775 | 228 | 135 | 398 | 1158 | 126 | 304 | 950 |
| FDSGGRU | 301 | 135 | 291 | 50 | 1733 | 299 | 692 | 201 | 128 | 418 | 1174 | 123 | 252 | 903 |
| ASGGRU | 311 | 133 | 266 | 54 | 1663 | 267 | 646 | 189 | 110 | 331 | 911 | 116 | 239 | 806 |
| CEEMDAN-ASGGRU | 287 | 112 | 223 | 37 | 684 | 278 | 544 | 180 | 122 | 325 | 638 | 103 | 218 | 826 |
| LCEEMDAN-ASGGRU | 191 | 91 | 183 | 33 | 744 | 258 | 387 | 158 | 97 | 267 | 556 | 87 | 128 | 518 |





Table 3 NSE of the six models at 14 stations (S1–S14), averaged over 10 independent runs.

| Model | NSE | | | | | | | | | | | | | |
|---|---|---|---|---|---|---|---|---|---|---|---|---|---|---|
| | S1 | S2 | S3 | S4 | S5 | S6 | S7 | S8 | S9 | S10 | S11 | S12 | S13 | S14 |
| LSTM | 0.579 | 0.654 | 0.662 | 0.647 | 0.838 | 0.709 | 0.794 | 0.672 | 0.742 | 0.733 | 0.808 | 0.544 | 0.760 | 0.796 |
| DTWSGGRU | 0.520 | 0.706 | 0.689 | 0.622 | 0.818 | 0.713 | 0.804 | 0.694 | 0.749 | 0.813 | 0.803 | 0.549 | 0.712 | 0.835 |
| FDSGGRU | 0.608 | 0.732 | 0.706 | 0.734 | 0.821 | 0.704 | 0.845 | 0.762 | 0.775 | 0.792 | 0.801 | 0.574 | 0.804 | 0.852 |
| ASGGRU | 0.580 | 0.735 | 0.755 | 0.691 | 0.839 | 0.765 | 0.865 | 0.789 | 0.835 | 0.870 | 0.879 | 0.616 | 0.823 | 0.881 |
| CEEMDAN-ASGGRU | 0.641 | 0.814 | 0.827 | 0.849 | 0.971 | 0.742 | 0.902 | 0.807 | 0.796 | 0.871 | 0.938 | 0.695 | 0.841 | 0.867 |
| LCEEMDAN-ASGGRU | 0.837 | 0.868 | 0.883 | 0.883 | 0.967 | 0.779 | 0.951 | 0.851 | 0.870 | 0.907 | 0.954 | 0.785 | 0.949 | 0.948 |





Table 4 Learned adaptive adjacency matrix with the flow-direction (FD) and DTW-based graphs under varying threshold levels.

| Threshold | An Edge Count | FD-A Overlap | DTW-A Overlap | FD-only Edges | DTW-only Edges | A-only Edges |
|---|---|---|---|---|---|---|
| 0.05 | 72 | 80.5% | 32.1% | 0 | 144 | 0 |
| 0.10 | 57 | 78.0% | 25.4% | 0 | 158 | 0 |
| 0.20 | 45 | 70.7% | 20.1% | 0 | 167 | 0 |