# Peer review of "A hybrid model for streamflow prediction addressing spatial connectivity and non-stationary dynamics with adaptive graph learning and multiscale decomposition"

_EGUsphere, 2025_

## Author Comment (AC2)

Dear Anonymous Referee,

We sincerely thank the reviewers for their constructive and insightful comments. Your feedback has substantially improved the quality, clarity, and organization of our work.

Comments from Reviewer #1:

This study proposes a hybrid model (LCEEMDAN-ASGGRU) integrating logarithmic transformation, CEEMDAN decomposition, and adaptive graph neural network for streamflow forecasting. The research topic holds practical application value, with systematic empirical studies conducted in the Poyang Lake Basin. However, the paper has several areas requiring improvement in terms of innovation, experimental design, and technical depth. Major revisions are recommended before considering publication.

*Response:*

*We sincerely appreciate the reviewer's thoughtful and constructive comments. Below, we have provided responses to each comment and described how the suggested changes have been incorporated into the revised manuscript.*

My comments are provided as follows:

Major Comments

1. The combination of CEEMDAN with deep learning has been explored in recent literature (e.g., Xu et al., 2024). Furthermore, logarithmic transformation represents a standard preprocessing technique that cannot be considered a primary contribution. The authors should clearly articulate what distinguishes their approach from existing methods and explicitly states the novel aspects of the proposed framework.

*Response:*

*Thanks for your comment. We agree that both CEEMDAN and the logarithmic transform are established techniques, and that CEEMDAN-deep learning hybrids (Xu et al., 2024) have increasingly been applied in streamflow prediction. In the revised manuscript, we have tempered the novelty claims in the Introduction and contribution statement, and we now explicitly state the logarithmic transformation as a standard stabilizing preprocessing step rather than a primary contribution.*

*Our intention is not to present CEEMDAN or log-based preprocessing as a new technique, but to investigate how multiscale decomposition and graph neural networks interact in a multi-station setting, where both spatial connectivity and non-stationary dynamics are important for streamflow forecasting. To the best of our knowledge, most existing CEEMDAN-based hydrological studies couple CEEMDAN with LSTM/GRU/CNN at the level and do not explicitly encode spatial dependence among stations (e.g. Ghanbari-Adivi and Ehteram, 2025; Li et al., 2023), while GNN-based hydrological studies typically rely on graphs when representing spatial relationships across multiple stations.*

*In this respect, our contribution lies at the level of an integrated framework and its systematic evaluation. Specifically, the proposed model couples CEEMDAN's multi-scale temporal decomposition with an adaptive graph recurrent architecture, enabling the extraction of the intrinsic temporal characteristics of streamflow at different frequencies and their spatial propagation patterns across the watershed network.*

*In addition to proposing an integrated forecasting pipeline, the manuscript also aims to provide interpretive insights into spatial dependencies and predictive uncertainty. First, we analyze the adaptive adjacency matrix learned by ASGGRU on the original series (Section 4.4, Fig. 7), and show*

*that many strong directed edges coincide with known upstream-downstream relationships, while others reflect cross-basin connections plausibly driven by shared meteorological forcing. This indicates that learned graph captures hydrologically meaningful connectivity without explicitly encoding river topology. Second, each deterministic model is coupled with an HMM-GMR post-processor to quantify forecast uncertainty and evaluate coverage, sharpness and CRPS across models (Section 4.6). This provides a systematic assessment of uncertainty for the LCEEMDAN-ASGGRU hybrid framework and helps interpret performance gains.*

*Overall, we have revised the manuscript to temper the novelty claims around CEEMDAN and log-transform preprocessing and to more clearly articulate the contribution of combining multiscale decomposition, adaptive graph learning, and uncertainty quantification for multi-station streamflow forecasting.*

[1] Ghanbari-Adivi, E., Ehteram, M., 2025. CEEMDAN-BILSTM-ANN and SVM models: two robust predictive models for predicting river flow. Water Resour. Manage. 39, 3235–3271. https://doi.org/10.1007/s11269-025-04105-w

[2] Li, H., Zhang, X., Sun, S., Wen, Y., Yin, Q., 2023. Daily flow prediction of the huayuankou hydrometeorological station based on the coupled CEEMDAN–SE–BiLSTM model. Sci. Rep. 13, 18915(2023). https://doi.org/10.1038/s41598-023-46264-z.

2. (Line 244) "Each variable (streamflow, precipitation, and temperature) at each station was decomposed into eight IMFs and one residual component." The manuscript states that each variable is decomposed into eight IMFs plus one residual component without providing theoretical or empirical justification. This parameter selection requires thorough discussion including sensitivity analysis of IMF numbers, and comparison with alternative decomposition levels.

*Response:*

*Thank you for the suggestion. In our implementation using the PyEMD library's CEEMDAN with its default parameters, the algorithm adaptively determined the number of IMFs, resulting in varying counts across different variables and stations. For our data, precipitation series decompose into 10 or 11 IMFs, streamflow into 9 or 10, and temperature into 8 or 9. Using these counts directly would make the input channel sizes inconsistent across variables/stations.*

*To align channels, we merge all components slower than IMF8 (i.e., IMFs with indices > 8) into a low-frequency aggregated residual. Thus, "8 IMFs + 1 residual" in the manuscript actually refers to "the first 8 CEEMDAN modes kept as separate channels, with all remaining modes aggregated into the residual". This preserves exact reconstruction while ensuring a consistent number of channels across variables and stations. We have clarified this description in the revised Section 3.1.*

*We acknowledge your point that this design choice warrants empirical justification. To this end, we conducted a sensitivity analysis in which the proposed LCEEMDAN-ASGGRU model was re-trained using K = 5, 6, and 7 IMFs (plus the aggregated residual), while keeping all other settings unchanged. The results are summarized in Table A1. Compared with the K = 8 setting used in the main manuscript, using K = 6 or K = 7 leads to lower predictive skill. The K = 5 and K = 8 configurations achieve similarly high mean NSE (0.889 ± 0.040 and 0.888 ± 0.012), but K = 8 yields the smallest RMSE (264 ± 23 m³/s) and the lowest run-to-run variability for both NSE and RMSE. This indicates that retaining eight IMFs offers a more accurate and robust representation, while still allowing higher-order modes to be safely aggregated into the residual. Consequently, we adopt K = 8 as the default decomposition level in the main experiments.*

Table A1: Comparison of LCEEMDAN-ASGGRU performance under different numbers of IMFs.

| The number of IMFs | 5 | 6 | 7 | 8 |
|---|---|---|---|---|
| NSE $\pm$ std | $0.889 \pm 0.040$ | $0.817 \pm 0.042$ | $0.857 \pm 0.063$ | $0.888 \pm 0.012$ |
| RMSE $\pm$ std (m³/s) | $275 \pm 73$ | $390 \pm 77$ | $299 \pm 70$ | $264 \pm 23$ |

*We have added a brief explanation of the decomposition-level choice in Section 3.1. In addition, we have created **Appendix A** in the revised manuscript, which provides the complete sensitivity analysis and the corresponding results in **Table A1**.*

3. In terms of the model design, LCEEMDAN-ASGGRU feeds nine decomposed components into ASGGRU individually to construct nine sub-models, with the final prediction result derived as the average of these sub-models. This design appears to overlook the interactivity among the decomposed components, rendering the approach more akin to an ensemble model rather than a truly integrated hybrid framework. It is therefore requested that the authors explain the rationale behind treating the nine sub-models as independent entities instead of exploring the interactive fusion of their training features. Why was the potential for synergistic information exchange between decomposed components not considered in the model architecture?

*Response:*

*Thank you for this thoughtful comment. We first clarify that, in our implementation, the final prediction is obtained by **summing**, not averaging, the outputs of the component-wise submodels. This summation is the inverse operation of the CEEMDAN decomposition and is the standard way to reconstruct the original signal from its components. We have revised Section 3.4 to make this reconstruction step explicit in both text and figure.*

*In LCEEMDAN-ASGGRU, nine scale-specific branches were constructed, each branch receives the k-th component (streamflow, precipitation and temperature at that scale, across all stations) as input, learns a scale-specific adaptive graph and temporal dynamics via ASGGRU, and outputs the predicted k-th component of streamflow. The final forecast is obtained by summing the predicted components. In this sense, the "nine sub-models" are **scale-conditioned modules** that mirror the CEEMDAN representation and allow the ASGGRU to specialize in different frequency bands.*

*Regarding the reviewer's suggestion on "exploring the interactive fusion of their training features", we have implemented and tested a cross-scale fusion design. Specifically, all CEEMDAN components (IMFs and residual) were concatenated along the feature dimension and provided them as joint input to ASGGRU, so that each prediction step had access to the full multiscale representation. All other settings were kept unchanged.*

*The results of this experiment are summarized in Table R1 of this response. The cross-scale fusion variant leads to a substantial degradation in performance: the mean NSE over the 14 stations drops to about 0.63, which is markedly lower than the NSE of the original LCEEMDAN-ASGGRU, and the corresponding RMSE is consistently higher. We have also observed that this fused-input design requires a much larger input dimension and parameter count, increasing training time without delivering accuracy gains. These findings suggest that, for the present dataset and model capacity, explicitly mixing all scales at the input level tends to blur the scale-specific structure extracted by CEEMDAN and does not improve predictive skill.*

Table R1: Comparison of predictive metrics between LCEEMDAN-ASGGRU and Cross-scale fusion

variant.

| | LCEEMDAN-ASGGRU | Cross-scale fusion variant |
| --- | --- | --- |
| NSE ± std | 0.888 ± 0.012 | 0.686 ± 0.070 |
| RMSE ± std (m³/s) | 264 ± 23 | 616 ± 95 |

Minor Issues

1. (Line 22) The phrase "mean root mean squared error of 264" lacks units, which is essential for RMSE (a dimensional metric). Please thoroughly review the entire manuscript and correct the notation for all dimensional units to ensure clarity and rigor.

*Response:*

*Agree. In the revised manuscript, we have added the physical unit "m³/s" wherever the root mean square error (RMSE) is reported. We have also carefully checked the entire manuscript, including the main text, tables, and figure captions, to ensure that dimensional quantities are consistently reported with their units.*

2. (Figure 1) Elevation values in the figure require associated units. Additionally, the manuscript mentions five major rivers (Ganjiang, Fuhe, Raohe, Xinjiang, and Xiushui) as key tributaries of the Poyang Lake Basin, but these rivers are not explicitly displayed in Figure 1. Given the statement that "Figure 1 shows the spatial distribution of selected stations, covering the five major river systems that drain into the Poyang Lake," the five corresponding sub-basins should be clearly labeled in the figure to align with the textual description.

*Response:*

*Thank you for your suggestions. We have revised Figure 1 as follows: (i) the elevation colour scale now includes units (m); (ii) the five major tributaries of the Poyang Lake basin (the Ganjiang, Fuhe, Raohe, Xinjiang, and Xiushui) have been clearly labelled.*

[Figure]

Figure 1: Location of hydrological stations.

3.    (Figure 3) The model framework diagram necessitates optimization for clarity: (1) The meaning of "T1-T4383" following normalization is not clarified, please add explicit annotations to define these variables; (2) Relocating this framework diagram to Section 3.1 (Methods) would better facilitate readers' understanding of the model structure prior to discussing results; (3) The arrow placement between the 4th, 5th, and 6th sub-diagrams is ambiguous and fails to illustrate the data transfer and computational flow from the upper to lower layers of the model; (4) The boxes in the last two sub-diagrams are incomplete, and the label "LCEEMDAN-ASGGRU" is misleading, which should refer to the prediction output rather than the model name.

*Response:*

*We thank the reviewer for these helpful suggestions regarding Figure 3. In the revised manuscript, Figure 3 has been redrawn to improve clarity and readability. The main changes are as follows:*

*(1) "T1–T4383" symbols represent the sequence of daily time steps in the full dataset. We have added an explicit explanation in the figure caption:*

*"The notation T1-T4383 represents the sequence of daily time steps in the full dataset (from day 1 to day 4383)."*

*(2) We updated the arrows to clearly illustrate the top-to-bottom flow of information. The revised figure now shows: (i) CEEMDAN outputs feeding into a "Normalization and sample construction" step; (ii) the construction of training and testing sequences (sliding windows T1-T12, ..., Tm+13-Tm+24, etc.); and (iii) these sequence samples being used as inputs to the ASGGRU sub-models. This addresses the previous ambiguity in how information is transferred from the upper preprocessing steps to the lower model layers.*

*(3) The label in the final sub-diagram has been changed from "LCEEMDAN-ASGGRU" to "Prediction*

*output" to correctly refer to the model output rather than the model name.*

[Figure]

Figure 3: Framework of LCEEMDAN-ASGGRU. The notation T1-T4383 represents the sequence of daily time steps in the full dataset (from day 1 to day 4383).

*We carefully considered the suggestion of moving this framework diagram to Section 3.1. Section 3.1 only introduces the CEEMDAN decomposition, whereas Section 3.4 describes the complete framework. For this reason, we keep Figure 3 in Section 3.4, where the full framework is first presented.*

4.    (Sections 4.1–4.2) The content in these sections focuses on experimental design and comparative model settings, which is misclassified under the Results section. It is recommended to create a dedicated chapter (e.g., "Experimental Setup" or "Comparative Models") following the Methods section to house this material. Furthermore, the abbreviations DTWSGGRU and FDSGGRU are used without prior definition.

*Response:*

*Thank you for this helpful suggestion. We agree that the description of the experimental design and the comparative model settings fits more naturally within the methodological content rather than in the Results section.*

*In the revised manuscript, we have added a new subsection Section 3.7, titled "Experimental setup and comparative models". This subsection gathers the content from sections 4.1 and 4.2 of the original submission versions.*

*Within Section 3.7, all model abbreviations are now introduced and clearly defined. We have also checked the rest of the manuscript to ensure that all abbreviations are defined at their first occurrence.*

5.    (Section 4.5) The authors should explicitly justify the selection of stations S4 and S5 for detailed analysis, i.e., clarify their representativeness. Additionally, Figure 9 suffers from insufficient resolution, making its core message unintelligible.

*Response:*

*Thank you for the comment. In the original manuscript, L409-L414, we noted that S4 is located in the southernmost part of the basin and exhibits the lowest mean streamflow, while S5 is located in the*

*northernmost part and records the highest mean streamflow. Since all 14 stations are modelled by GNN jointly, model performance at these low-flow and high-flow extremes provides a stringent test of model robustness across the remaining stations.*

*In the revised manuscript, we have made the rationale for selecting S4 and S5 more explicit:*

*L409-L412: "To further visualize these improvements, Figure 6 presents the observed and predicted streamflow of six models at two 410 representative stations: S4, located in the southernmost part of the basin and exhibiting the lowest mean streamflow, and S5, located in the northernmost end and recording the highest mean streamflow. Together, they provide a stringent complement to the overall performance metrics summarized in Figure 4."*

*This revision clarifies why S4 and S5 are used for detailed visualization and highlights their representativeness with respect to both flow magnitude and network position.*

*Regarding **Figure 9**, we acknowledge that the original submission may have hindered interpretability. In the revised version, Figure 9 has been regenerated at substantially higher resolution, with enhanced line weights, larger font sizes, and clearer shading for the prediction intervals.*

[Figure]

Figure 9: Time-series plots of predicted means and 95% prediction intervals for the six models at stations S5 (a) and S4 (b).

*We have also revised the relevant descriptive text:*

*L481-L489:*

*"Figure 9 provides a time-series illustration of the probabilistic forecasts for stations S5 and S4, where the black line denotes the observed streamflow, the blue line represents the prediction mean (the*

*conditional expectation from the HMM-GMR predictive distribution), and the shaded area indicates the associated 95% prediction interval. This visualization allows a direct assessment of both point-forecast accuracy and the reliability of the uncertainty estimates.*

*At both stations, LCEEMDAN-ASGGRU exhibits the closest agreement between prediction means and observations while maintaining the narrowest and most responsive uncertainty bands among all six models. In the high-flow regime at S5 (Fig.9 (a)), the model tracks peak magnitudes without causing undue interval expansion, whereas in the low-flow regime at S4 (Fig.9 (b)), it captures subtle fluctuations with tight yet well-calibrated intervals, reflecting consistent coverage and stable predictive behaviour across contrasting hydrological conditions."*

6. (Line 433) "In particular, connections such as S3→S10, S4→S14, S7→S11, and S13→S7 reflect physically consistent upstream–downstream dependencies that are embedded in the true hydrological topology." However, the figure also shows S10→S3. Therefore, it is necessary to provide a detailed explanation of the adjacency matrix, clarifying that it encodes not only hydrological upstream–downstream relationships but also spatial meteorological correlations (attributed to the inclusion of precipitation and temperature data) to resolve this apparent contradiction.

*Response:*

*Thank you for the helpful comment. The adaptive adjacency matrix learned by ASGGRU is indeed constructed from **multi-variable inputs**, including streamflow, precipitation and temperature at all stations. It is not restricted to a purely one-way river-routing structure, but can encode both hydrological routing and spatially coherent meteorological signals.*

*In the revised manuscript, we clarify this interpretation in Section 4.4 and adopt more precise wording to describe how the adaptive adjacency matrix aligns with known hydrological connectivity.*

*We revised the description around L433:*

*"To investigate the spatial representations captured by the adaptive graph learning mechanism, we analyze the learned graph obtained from ASGGRU and evaluate its hydrological consistency and its relationship with static graph priors.*

*The learned adaptive adjacency matrix is visualized in Fig. 7. Several strong directional connections emerge from the learned structure, which are consistent with known flow-direction relationships within the river network. In particular, connections such as S3→S10, S4→S13, S7→S14 and S13→S7 consistent with physically plausible upstream-downstream dependencies inferred from the catchment topology. This indicates that the data-driven learning process is able to recover some key aspects of the underlying hydrological connectivity without any explicit spatial supervision. This result echoes findings by Bai and Tahmasebi (2023), who also observed that adaptive GNNs can learn spatial structures that align with physical understanding in groundwater forecasting and environmental modeling.*

*However, the adaptive adjacency matrix is learned jointly from streamflow, precipitation and temperature inputs. It also contains reciprocal and cross-basin links that go beyond a purely one-way river-routing graph, reflecting shared meteorological forcing or residual statistical dependence between stations. More generally, such indirect or cross-basin connections likely capture remote influences or common climatic drivers, complementing the river-network signal."*

7. The terms "streamflow" and "runoff" are mixed throughout the manuscript.

*Response:*

*Thank you, we standardized the terminology in the revision.*

8.    The standard deviations of the 10 repeated experiments should be reported.

*Response:*

*Thank you for the suggestion. We have now reported the run-to-run standard deviations of the 10 repeated experiments for both NSE and RMSE. The results are summarized at the model level (mean ± standard deviation) and provided in **Appendix A as Table A2**. This table quantifies variability across independent runs and demonstrates that the proposed LCEEMDAN-ASGGRU maintains low run-to-run variability while achieving the best overall accuracy among the six models.*

Table A2: Overall model performance (mean ± standard deviation) across 10 repeated runs.

| Model | NSE ± std | RMSE ± std(m³/s) |
|---|---|---|
| LSTM | 0.710 ± 0.017 | 503 ± 14 |
| DTWASGGRU | 0.716 ± 0.017 | 496 ± 50 |
| FDASGGRU | 0.751 ± 0.061 | 479 ± 55 |
| ASGGRU | 0.779 ± 0.015 | 432 ± 28 |
| CEEMDAN-ASGGRU | 0.826 ± 0.036 | 327 ± 43 |
| LCEEMDAN-ASGGRU | 0.888 ± 0.012 | 264 ± 23 |

---

## Author Comment (AC3)

Dear Anonymous Referee:

We are grateful for your thoughtful and constructive comments. Your feedback has significantly strengthened the manuscript in terms of clarity, methodological soundness, and presentation. We have provided detailed responses to each comment.

**Overall comment**

This paper addresses the problem of daily streamflow forecasting of multi-station by applying CEEMDAN decomposition to extract multiscale dynamic features, and an adaptive graph recurrent network to capture spatiotemporal dependencies. The authors also discuss the advantages of uncertainty-based interpretation. Overall, the manuscript is clearly written and generally well organized. However, the proposed approach mainly combines existing techniques rather than introducing a truly novel methodology. The framework—decomposing the time series, modeling each component separately, and then reconstructing the final output—is quite common in the literature. Moreover, the model appears to focus on only one-step-ahead forecasting, which limits its practical value for real-world hydrological applications. Therefore, I don't recommend the publication of the manuscript in HESS in its present form. *Response:*

*We sincerely thank the reviewer for the positive assessment regarding the clarity and organization of the manuscript, as well as for the constructive concerns about methodological novelty and forecasting horizon. We agree that CEEMDAN and the logarithmic transform are established techniques, and that "decompose-model-recompose" frameworks have been widely adopted in hydrological forecasting studies. In the revised manuscript, we have adjusted the Introduction and contribution statements to avoid overstating the novelty of these individual components.*

*Our intention is not to claim that CEEMDAN or log-based preprocessing as methodological innovations. Instead, our focus is on how multiscale decomposition and graph neural networks interact in a multi-station setting where both spatial connectivity and non-stationary dynamics play central roles. To the best of our knowledge, most existing CEEMDAN-based hydrological studies couple CEEMDAN with LSTM/GRU/CNN at the single-station level, while GNN-based hydrological studies often relies on static graphs when representing spatial relationships among multiple stations. In this respect, our contribution lies at the level of an integrated framework and its systematic evaluation. Specifically, the proposed model couples CEEMDAN's multi-scale temporal decomposition with an adaptive graph*

*recurrent architecture, enabling the extraction of the intrinsic temporal characteristics of runoff at different frequencies and their spatial propagation patterns across the watershed network.*

*In addition to proposing an integrated forecasting pipeline, the manuscript also aims to provide interpretive insights into spatial dependencies and predictive uncertainty. First, we analyze the adaptive adjacency matrix learned by ASGGRU on the original series (Section 4.4, Fig. 7), and show that many strong directed edges coincide with known upstream-downstream relationships, while others reflect cross-basin connections plausibly driven by shared meteorological forcing. This indicates that learned graph captures hydrologically meaningful connectivity without explicitly encoding river topology. Second, each deterministic model is coupled with an HMM-GMR post-processor to quantify forecast uncertainty and evaluate coverage, sharpness and CRPS across models (Section 4.6). This provides a systematic assessment of uncertainty for the LCEEMDAN-ASGGRU hybrid framework at the basin scale and helps interpret performance gains.*

*Regarding the reviewer's concern that "one-step-ahead forecasting limits model's practical value for real-world hydrological applications", we acknowledge that the main experiments in the current version focus on one-day-ahead daily prediction (Section 4.1). To address this concern, we have conducted additional experiments where separate instances of the proposed LCEEMDAN-ASGGRU model were trained for lead times from 2 to 7 days ahead, using the same 12-day input window. The results are included in the revised manuscript. Figure R1 in this response summarizes the mean NSE over the 14 stations for lead times from 1 to 7 days, the mean NSE decreases smoothly from 0.888 at lead-1 to 0.832, 0.767, 0.762, 0.728, 0.695 and 0.662 at lead-2 to lead-7, respectively. This behavior is consistent with the expected degradation of forecast skill as the lead time increases, and indicates that the proposed architecture still provides useful predictive information several days in advance.*

[Figure]

*Figure R1: The NSE of different lead time.*

In summary, we have revised the manuscript to:

(i) temper the novelty claims around CEEMDAN and log-transform preprocessing;

(ii) more clearly articulate the contribution of combining multiscale decomposition, adaptive graph learning and uncertainty quantification for multi-station streamflow forecasting, including an illustrative spatial analysis of the adaptive graph learned by ASGGRU;

(iii) address the original focus on one-day-ahead prediction by incorporating comprehensive multi-day (2-7 day) lead-time experiments.

**Specific comments**

L120: Figure 1 is not clear, and this issue persists throughout the manuscript. Figures in general need to be improved in clarity and readability.

*Response:*

*Thank you for the comment. We revised Figure 1 by adding an elevation color scale with units (m) and explicitly labelling the five major tributaries of the Poyang Lake Basin (Ganjiang, Fuhe, Raohe, Xinjiang, and Xiushui). In addition, we have improved the clarity of all figures throughout the manuscript by increasing resolution, enhancing line and marker weights, and enlarging font sizes. These adjustments collectively make the figures clearer and more readable.*

[Figure]

*Figure 1 : Location of hydrological stations.*

L202: The abbreviation Aadp is not defined.

*Response:*

*Thank you for pointing this out. We have supplemented the definition of $A_{adp}$ in Section 3.2.*

*L203 below Eq. (6) has been revised to read:*

*"where $A_{adp} \in R^{N \times N}$ denotes the adaptive adjacency matrix, $E_1, E_2 \in R^{N \times e}$ are two trainable node-embedding matrices, $N$ is the number of stations and $e$ the embedding dimension."*

L208: The meaning of arrow in the lower right corner of Figure 2 is unclear.

*Response:*

*Thank you. We improved all arrows for readability.*

L212: Is SGGRU proposed by the authors' team, or is it based on Zhao et al. (2020)? Please clarify.

*Response:*

*Thank you for the question. Our spatial graph gated recurrent unit (SGGRU) follows the same principle as the T-GCN cell proposed by Zhao et al. (2020), in which graph convolution is embedded within a*

*GRU to jointly model spatial and temporal dependencies. Zhao et al. (2020) implement their model using an undirected road-network graph, whereas in our study the framework is instantiated with directed and potentially asymmetric graphs.*

*To eliminate ambiguity, we revised the description of SGGRU in Section 3.3:*

*Line 210: "Following the temporal graph convolutional unit of Zhao et al. (2020), we adopt a GRU cell in which the affine transformations are replaced by graph-convolution operators defined on either fixed or learned adjacency matrices. We refer to this generic graph-convolutional GRU as the spatial graph gated recurrent unit (SGGRU) in this study."*

L257: Corresponding to the logarithmic and standardization steps in Step 1 and 3, shouldn't inverse standardization and inverse log-transformation be applied before combining the final results? (Figure 3 indicates such steps are needed.)

*Response:*

*We thank the reviewer for raising this point. The reviewer is correct that the preprocessing in Steps 1 and 3 (logarithmic transform and standardization) must be inverted when reconstructing the final streamflow series. This is indeed how the model is implemented, but our description in Section 3.4 and Figure 3 was not sufficiently explicit. We have therefore rewritten this part of the text and clarified the full pipeline as follows , with Figure 3 revised to reflect these clarifications:*

[Figure]

Figure 3: Framework of LCEEMDAN-ASGGRU. The notation T1-T4383 represents the sequence of daily time steps in the full dataset (from day 1 to day 4383).

*Line235:*

*(1)   To stabilize variance and mitigate the influence of extreme values observed in streamflow and precipitation time series, we apply a logarithmic transformation prior to decomposition. Specifically, we adopt the natural logarithm with the transformation defined as:*

$$X_{\log}(t) = \log(1 + X(t)) \tag{10}$$

*This transformation ensures numerical stability while reducing the influence of large outliers, thereby enhancing the subsequent CEEMDAN decomposition and improving the separability of IMFs across scales.*

*(2)   The CEEMDAN decomposition was performed using the PyEMD library with default parameters, including a noise standard deviation of 0.2 and 250 noise-assisted trials. Each variable (streamflow, precipitation, and temperature) at each station was decomposed into eight IMFs and one residual component.*

*(3)   Each IMF and residual component is standardized using z-score normalization before forecasting to ensure they remain on the same scale:*

$$X_k^{'}(t) = \frac{X_{\log,k}(t) - \overline{X}_{\log,k}}{\sigma_k} \tag{11}$$

*where $X_k^{'}(t)$ denotes the normalized series of the k-th component, $X_{\log,k}(t)$ denotes the k-th IMF in log domain. $\overline{X}_{\log,k}$ and $\sigma_k$ denote the mean and standard deviation of this component. These two corresponding normalization parameters were stored and applied during postprocessing to enable accurate inverse transformation of the model predictions back to the original scale.*

*(4)   Each of the nine decomposed components from the CEEMDAN process is treated as an independent prediction subtask. For each component, a separate instance of the ASGGRU model is trained independently, allowing the model to specialize in capturing the temporal dynamics unique to that frequency scale. Notably, each IMF and the residual has its own set of optimal hyperparameters, reflecting the varying statistical characteristics and predictive complexities across components. This design provides additional flexibility, enabling the model to allocate capacity appropriately.*

*(5)   During inference, each trained submodel outputs a predicted sequence corresponding to its specific component. These component outputs are first mapped back to the log domain via inverse z-score normalization, then linearly aggregated across all components to reconstruct the log-transformed*

*streamflow series, and finally converted to the original streamflow scale using the inverse log-transform. Mathematically, the reconstruction can be expressed as:*

$$y'_{\log,k}(t) = y'_k(t)\sigma_k + \bar{X}_{\log,k}(t) \tag{12}$$

$$\hat{y}_{\log}(t) = \sum_{k=1}^{K} y'_{\log,k}(t) \tag{13}$$

$$\hat{y}(t) = \exp(\hat{y}_{\log}(t)) - 1 \tag{14}$$

*where $y'_k(t)$ denotes the prediction of the k-th IMF or residual submodel in the ASGGRU submodel. $y'_{\log,k}(t)$ is the inverse-normalized value. $\hat{y}_{\log}(t)$ denotes the reconstructed streamflow series in log domain, and $\hat{y}(t)$ is the final streamflow prediction value after applying the inverse logarithmic transformation.*

L360: What does the learned graph structure of ASGGRU look like after training? Please provide a visualization. Which also makes me confused in L436.

*Response:*

*Thank you for raising this point. In the submitted manuscript, the adaptive adjacency matrix learned by ASGGRU is already visualized as a heat map in Section 4.4 (Figure 7). However, we realize that this figure was only introduced later in Section 4.4, the discussion around L360 and L436 may have been difficult to follow on first reading.*

[Figure]

Figure 7: Learned adaptive adjacency matrix obtained from ASGGRU.

*In the revised manuscript, we have clarified connection. When introducing ASGGRU and its adaptive graph mechanism in Section 4.3.2, we now explicitly refer the reader to the spatial analysis in Section 4.4 and Figure 7, where the learned adaptive graph is visualized and interpreted.*

*Specifically, Line 370 has been revised to:*

*"In contrast, the ASGGRU model learns an adaptive spatial graph via node embeddings that are updated jointly with the model parameters, the spatial patterns of the learned adaptive graph are further analyzed in Section 4.4 (see Figure 7)."*

L421: From Figure 6, the ASGGRU results exhibit notable under- and over-estimations of flood peaks, and perform no better than DTWSGGRU and FDSGGRU. This appears inconsistent with the performance metrics reported in the text. Could the authors provide additional metrics, specifically for high-flow and low-flow conditions, based on Figure 4?

*Response:*

*Thank you for the comment. Figure 6 displays time series at two representative stations (S4 and S5), whereas the performance metrics summarized in Figure 4 are aggregated over all 14 stations and over the whole flow range. We agree that when focusing specifically on the flood peaks in Figure 6, the*

ASGGRU-based curves still display instances of underestimation and overestimation during some extreme events. This may appear inconsistent with the overall NSE values.

To examine this more systematically, we conducted an additional evaluation in which model performance is stratified by flow condition. For each station, the empirical Q75 of the observed daily streamflow was used as the threshold. Days with $Q \geq Q75$ were classified as high flow and days with $Q < Q75$ as low-to-moderate flow. NSE and RMSE were then computed separately for the two subsets and averaged over the 14 stations and 10 runs. The resulting performance metrics for high flow and low-to-moderate flow NSE and RMSE are summarized in Figure R2 of this response. This figure and corresponding analyzes have been provided in the revised manuscript.

Considering both conditions together, the Q75-based analysis shows a consistent ranking of model performance. Under high-flow conditions ($Q \geq Q75$), LCEEMDAN-ASGGRU attains the best skill among all models, with higher NSE and lower RMSE than CEEMDAN-ASGGRU, ASGGRU with the learned adaptive graph, the two static-graph (DTWSGGRU and FDSGGRU), and LSTM. In this condition, the ordering of high flow NSE aligns well with the basin-wide NSE reported in Figure 4, indicating that the overall metrics already reflects relatively high-flow performance.

For low-to-moderate flow condition ($Q < Q75$), the differences between models are even more pronounced. LCEEMDAN-ASGGRU again shows the strongest performance, while the static-graph models and the CEEMDAN-ASGGRU (without log transform) exhibit substantially lower. ASGGRU maintains positive NSE in this condition and outperforms DTWSGGRU, FDSGGRU, and LSTM. Importantly, introducing the logarithmic transform in LCEEMDAN-ASGGRU yields a substantial improvement in low-to-moderate performance, highlighting its effectiveness in stabilizing errors under low-to-moderate conditions.

In summarize,because NSE is more sensitive to errors during high-flow periods, the mean NSE in Figure 4 already captures much of the high-flow model skill. The Q75-stratified analysis confirms the overall performance ranking remains robust across flow conditions, and the apparent peak mismatches in Figure 6 represent local examples rather than contradictions to the basin-wide metrics.

We also identified and corrected a labelling error in the caption of Figure 6 in the original submission (the order of S4 and S5 was inadvertently swapped). The revised caption now reads:

*"Figure 6: Observed and predicted streamflow at two stations: (a) S5, situated at the northernmost point and exhibiting frequent and extreme high-flow events; (b) S4, located at the southernmost point of the basin and characterized by persistently low-flow conditions."*

[Figure]

*Figure R2: Model performance comparison in high-flow and low-to-moderate conditions across different forecasting models.*

L446: The overlap ratio between the ASG-based graph and the flow-direction-based graph (FD) has been provided. Accordingly, the overlap ratio between the DTW-based graph (DTW) and the FD should be reported for comparison. The overlap ratio between the DTW-based graph (DTW) and the flow-direction-based graph (FD) should be quantified and reported.

*Response:*

*Thank you for this helpful suggestion. In the revised manuscript, we have extended the overlap analysis to explicitly include the DTW-based graph (DTW) and the flow-direction-based graph (FD).*

*We now report directional overlap ratios among the learned adaptive graph $A_{adp}$, the DTW graph and the FD graph. For two edge sets X and Y, we define the directional overlap X→Y as the fraction of edges in Y that also appear in X, that is, "X→Y = (number of edges shared by X and Y) / (number of edges in Y)".*

*For the adaptive graph $A_{adp}$, we focus on $A_{adp}$→FD and $A_{adp}$→DTW, which quantify how much of the physical-flow and similarity-based prior information is recovered by the learned graph. For the two static graphs, we additionally report DTW→FD and FD→DTW to characterize the extent to which flow-direction structures are embedded within the broader DTW similarity network.*

*During this recomputation, we identified a minor preprocessing error in our original script: a header row had been mistakenly interpreted as an additional node when constructing the adjacency matrices for overlap analysis. This issue affected only the previously reported numerical values and did not influence any part of the model training or prediction. We have corrected the error, recalculated all overlap ratios, and updated the relevant text and Table 4 in Section 4.4. The qualitative conclusions remain unchanged.*

*The updated results can be summarized as follows (see Table 4).*

*Table 4: Learned adaptive adjacency matrix with the flow-direction (FD) and DTW-based graphs under varying threshold levels.*

| Threshold | $A_{adp}$ edge counts | FD edge counts | DTW edge counts | $A_{adp}$→FD | $A_{adp}$→DTW | DTW→FD | FD→DTW | $A_{adp}$-only | DTW-only | FD-only |
|---|---|---|---|---|---|---|---|---|---|---|
| 0.05 | 33 | 13 | 182 | 38.5% | 18.1% | 100% | 7.1% | 0 | 119 | 0 |
| 0.1 | 18 | 13 | 182 | 30.8% | 9.9% | 100% | 7.1% | 0 | 132 | 0 |
| 0.2 | 6 | 13 | 182 | 7.7% | 3.3% | 100% | 7.1% | 0 | 139 | 0 |

*For all thresholds, the overlap between DTW and FD is 100 %, which means that every flow-direction edge is contained in the DTW graph. In other words, the physical flow-direction network is fully embedded within the DTW-based similarity graph, while FD links account for only a small fraction of all DTW edges (FD→DTW = 7.1%). The adaptive graph $A_{adp}$ can be viewed as a sparse, task-oriented subgraph extracted from this dense similarity network. It retains only a small proportion of DTW edges, and among those, it tends to preserve a larger share of flow-direction consistent links. At the 0.05 threshold, for example, A contains 33 effective edges, recovering 38.5 % of the FD but only 18.1 % of the DTW links. This pattern suggests that the learned graph selectively preserves temporally similar connections that are also hydrologically plausible, rather than inheriting the majority of DTW-only correlations.*

*The corresponding paragraph in Section 4.4 has been revised as follows:*

*"To further quantify the relationship between the learned adaptive graph and the two static priors, we conducted an overlap analysis under varying edge-weight thresholds (0.05, 0.10, and 0.20). For two edge sets X and Y, the directional overlap X→Y is defined as the fraction of edges in Y that also appear in X. For the adaptive graph $A_{adp}$, we report $A_{adp}$→FD and $A_{adp}$→DTW, which indicate how much of the physical-flow and similarity-based prior information is recovered by the learned graph. For the two prior graphs themselves (i.e., the static graphs), we report DTW→FD and FD→DTW to characterize how the FD structure is embedded within the broader DTW similarity network. The resulting overlap ratios and edge counts are summarized in Table 4.*

*The results reveal two main patterns. First, DTW→FD is 100 % for all thresholds, which means that every flow-direction edge is contained in the DTW graph. FD→DTW is 7.1% for all thresholds, which means that FD links account for only a small fraction of all DTW edges. The adaptive graph $A_{adp}$ can be viewed as a sparse, task-oriented subgraph extracted from this dense similarity network. At the 0.05 threshold, $A_{adp}$ contains 33 effective edges, which together recover 38.5 % of the FD links but only 18.1 % of the DTW links. As the edge-weight threshold is increased from 0.05 to 0.20, the number of retained edges in A decreases from 33 to 18 and 6, and the coverage of both FD and DTW links is reduced ($A_{adp}$→FD from 38.5% to 7.7 %, $A_{adp}$→DTW from 18.1 % to 3.3 %). Across all examined thresholds, the fraction of FD links recovered by $A_{adp}$ remains larger than that of DTW links, confirming that the learned graph is more strongly aligned with the flow-direction prior than with the full DTW similarity network. Importantly, no $A_{adp}$-only or FD-only edges are observed for any threshold, whereas a large number of DTW-only edges remain. This suggests that the adaptive graph learning primarily operates within the joint subspace spanned by the flow-direction and DTW priors, refining and re-weighting hydrologically and temporally meaningful spatial relationships rather than inventing entirely new connections."*

L460: According to the setup (3 × 2 × 2 = 12 models + baseline = 13), only six models are presented, please clarify the rationale.

*Response:*

*Thank you. In the original text, the* design space *was described as involving three spatial graphs (flow-direction, DTW, adaptive graph), CEEMDAN, and logarithmic transform, along with an LSTM*

baseline. We agree that this wording may have unintentionally suggested that a full $3 \times 2 \times 2$ factorial experiment was both implemented and fully reported. This was not our intention.

As state in L338-L341, our study follows a **hierarchical ablation strategy**, rather than an exhaustive enumeration of all 12 combinations:

"To validate the effectiveness of the proposed LCEEMDAN-ASGGRU, this section presents a comprehensive performance analysis through comparative experiments. Instead of evaluating each model in isolation, we adopt an ablation approach, progressively breaking down the proposed model into its key components: the adaptive graph learning module, the multi-scale CEEMDAN decomposition, and the logarithmic transformation."

***(i) Graph structure without decomposition or log transform.***

We first fix the preprocessing (no CEEMDAN, no logarithmic transform) and compare four architectures: LSTM, DTWSGGRU, FDSGGRU, and ASGGRU. This isolates the effect of spatial graph construction under the same temporal pipeline (SGGRU). As reported in Fig. 4 and Tables 2-3, all graph-based models outperform LSTM, and ASGGRU yields the best performance;

***(ii) Decomposition and log transform, conditional on the best graph.***

Next, we fix the spatial graph to the best-performing configuration (ASGGRU) and evaluate the contribution of CEEMDAN and the logarithmic transform. To this end, we report CEEMDAN-ASGGRU and LCEEMDAN-ASGGRU. With in this comparison, CEEMDAN-ASGGRU improves upon ASGGRU, and LCEEMDAN-ASGGRU further improves upon CEEMDAN-ASGGRU. The six models shown in the main text are correspond to this stepwise ablation and are validate the performance advantages of the proposed LCEEMDAN-ASGGRU.

Thus, the hierarchical ablation design clearly demonstrates the incremental contribution of each component and avoids unnecessary repetition.